# Activating SRC/MAPK signaling via 5-HT1A receptor contributes to the effect of vilazodone on improving thrombocytopenia

Ling Zhou[1†], Chengyang Ni[1†], Ruixue Liao[1†], Xiaoqin Tang[1], Taian Yi[2], Mei Ran[1,3], Miao Huang[2], Rui Liao[1], Xiaogang Zhou[1], Dalian Qin[1], Long Wang[1], Feihong Huang[1], Xiang Xie[4], Ying Wan[3], Jiesi Luo[3*], Yiwei Wang[3*], Jianming Wu[1,3,5*]

[1]Sichuan Key Medical Laboratory of New Drug Discovery and Druggability, Luzhou Key Laboratory of Activity Screening and Druggability Evaluation for Chinese Materia Medica, School of Pharmacy, Southwest Medical University, LuZhou, China; [2]School of Pharmacy, Chengdu University of Traditional Chinese Medicine, Chengdu, China; [3]School of Basic Medical Sciences, Southwest Medical University, Luzhou, China; [4]School of Basic Medical Sciences, Public Center of Experimental Technology, Model Animal and Human Disease Research of Luzhou Key Laboratory, Southwest Medical University, Luzhou, China; [5]Education Ministry Key Laboratory of Medical Electrophysiology, Southwest Medical University, Luzhou, China

*For correspondence:
ljs@swmu.edu.cn (JL);
wangyiwei0102@swmu.edu.cn
(YW);
jianmingwu@swmu.edu.cn (JW)

†These authors contributed equally to this work

**Abstract** Thrombocytopenia caused by long-term radiotherapy and chemotherapy exists in cancer treatment. Previous research demonstrates that 5-Hydroxtrayptamine (5-HT) and its receptors induce the formation of megakaryocytes (MKs) and platelets. However, the relationships between 5-HT1A receptor (5-HTR1A) and MKs is unclear so far. We screened and investigated the mechanism of vilazodone as a 5-HTR1A partial agonist in promoting MK differentiation and evaluated its therapeutic effect in thrombocytopenia. We employed a drug screening model based on machine learning (ML) to screen the megakaryocytopoiesis activity of Vilazodone (VLZ). The effects of VLZ on megakaryocytopoiesis were verified in HEL and Meg-01 cells. Tg (itga2b: eGFP) zebrafish was performed to analyze the alterations in thrombopoiesis. Moreover, we established a thrombocytopenia mice model to investigate how VLZ administration accelerates platelet recovery and function. We carried out network pharmacology, Western blot, and immunofluorescence to demonstrate the potential targets and pathway of VLZ. VLZ has been predicted to have a potential biological action. Meanwhile, VLZ administration promotes MK differentiation and thrombopoiesis in cells and zebrafish models. Progressive experiments showed that VLZ has a potential therapeutic effect on radiation-induced thrombocytopenia in vivo. The network pharmacology and associated mechanism study indicated that SRC and MAPK signaling are both involved in the processes of megakaryopoiesis facilitated by VLZ. Furthermore, the expression of 5-HTR1A during megakaryocyte differentiation is closely related to the activation of SRC and MAPK. Our findings demonstrated that the expression of 5-HTR1A on MK, VLZ could bind to the 5-HTR1A receptor and further regulate the SRC/MAPK signaling pathway to facilitate megakaryocyte differentiation and platelet production, which provides new insights into the alternative therapeutic options for thrombocytopenia.

## eLife assessment

This study presents a rather **valuable** finding that vilazodone can restore the normal platelet level through regulating 5-HT1A receptor. The evidence supporting the claims of the authors is **solid**, although inclusion of more cell lines and more detailed analysis of the results would have strengthened the study. The work will be of interest to scientists working in the field of thrombocytopenia.

## Introduction

5-HT has been identified as a vasoactive neurotransmitter mostly located in the periphery of the body, and less than 1% of total body serotonin circulates in its free form in the blood (*Erspamer and Asero, 1952*). Peripheral synthesis by enteric chromaffin cells and release into the blood are taken up by circulating platelets via serotonin transporters (SERTs) and stored in dense particles (*Spohn and Mawe, 2017*). In the blood, approximately 98% of 5-HT resides in platelets, which are released into circulation when activated. The biological effects of 5-HT are exerted through activation of any of 15 distinct receptors in 7 different classes 5-HT1 receptor to 5-HT7 receptor (5-HT1R to 5-HT7R) (*Barnes et al., 2021*). It has been assumed for a long time that 5-HT plays an important role in the hematopoietic process. 5-HT is a powerful mitogen that has been shown to have mitogenic action during embryonic development in mice (*Matsuda et al., 2004*). In particular, 5-HT is essential for the survival of hematopoietic stem cells and progenitor cells (HSPCs) in vivo throughout embryogenesis and increases the generation of HSPCs in vitro (*Lv et al., 2017*). Consistently, MKs were the first hematopoietic cells examined in relation to the serotonergic system and were megakaryocytes due to the well-known link between 5-HT and platelets. Studies have shown that the 5-HT2A, 5-HT2B, and 5-HT2C subtypes exist on the surface of human primary megakaryocyte progenitor cells, megakaryocytes, and some major megakaryocyte lines (*Yang et al., 2007*). In the process of megakaryocyte hematopoietic regulation, 5-HT increases the number of megakaryocyte progenitor cells, promotes megakaryocyte proliferation by activating the 5-HT2 receptor (5-HT2R), and promotes the formation and maturation of mouse CFU-MK and platelet generation in a dose-dependent manner (*Ye et al., 2014*). Additionally, 5-HT might contribute to erythropoiesis (*Amireault et al., 2011*). Overall, the serotonergic system is a therapeutic target that shows promise due to its extensive capacity for regeneration and regulation of hematopoietic lineages. In addition to 5-HT2R, the 5-HT1 receptor (5-HT1R) is also important for cell proliferation and differentiation, among which the 5-HT1A receptor (5-HTR1A) is one of the main subtypes involved. The presence of 5-HT1F receptor (5-HTR1F) on CD34+ cells suggested that 5-HT1R signaling may be involved in neuronal regulation of immature hematopoietic progenitors (*Skurikhin et al., 2001*). Moreover, 5-HT dose-dependently induces the migration of neural crest cells in culture through 5-HTR1A in mice to interfere with development (*Moiseiwitsch and Lauder, 1995*). Further evidence revealed that 5-HTR1A could exert effects on the phosphorylation of ERK1/2 (*Della Rocca et al., 1999*).

Vilazodone (VLZ) is a new dual-mechanism antidepressant approved by the United States Food and Drug Administration (FDA) for treating major depressive disorder (MDD). VLZ displays the property of an SSRI and also has partial activation of 5-HTR1A (*McCormack, 2015*). It selectively inhibits serotonin reuptake with a 50% inhibitory concentration (IC50) of 1.6 nmol/L and binds selectively to human 5-HT1A receptors with high affinity (IC50 of 2.1 nmol/L) (*Wang et al., 2015*). The antidepressant effect of VLZ is through inhibiting the serotonin transporter's reuptake of 5-HT, while part of the stimulant effect of 5-HTR1A is directed at the auto receptor-mediated negative neuronal feedback mechanism. Previous studies have shown that the 5-HTR1A partial activation of VLZ has a faster and larger effect on presynaptic 5-HTR1A (*Wang et al., 2015*). However, it is still unknown whether VLZ can regulate the fate of hematopoietic cells through 5-HTR1A.

Platelets are small nonnucleated cellular fragments that are essential for functions such as hemostasis, wound healing, angiogenesis, inflammation, and innate immunity (*Machlus and Italiano, 2013*). They are created in the bone marrow by megakaryocytes, which are enormous polyploid cells that develop from hematopoietic stem cells by megakaryopoiesis. The polyploid cells produced by this intricate process, which are remarkably similar in mice and humans, have a single polylobate nucleus and undergo repeated cycles of endomitosis after numerous rounds of unsuccessful cytokinesis (*Marini et al., 2022*). A low platelet count is a frequently encountered hematological abnormality. Thrombocytopenia is a medical condition associated with a reduced peripheral blood platelet count, which is characterized as a platelet count below $150\times10^9$ /L of whole blood (*Provan and Semple, 2022*).

Platelet production can be affected by bone marrow suppression, which is frequently observed in diseases including leukemia, chemotherapy, and sepsis (*Lien et al., 2021*). Among them, radiotherapy and chemotherapy for tumor treatment are the key factors leading to thrombocytopenia complications. Radiation-induced thrombocytopenia (RIT) is a significant cause of morbidity and mortality and is particularly important in the hematopoietic form of acute radiation syndrome (*Tkaczynski et al., 2018*). According to several studies, platelet count correlates more strongly with survival following total body irradiation than any other hematologic parameter (*Mouthon et al., 1999*; *Kiang et al., 2014*; *Lebois et al., 2016*). Bleeding and thrombocytopenia account for a large amount of morbidity and death in the context of radiation injury. Therefore, the creation of novel therapeutic approaches to manage thrombocytopenia is essential.

The continual development of platelet synthesis is known as megakaryocytopoiesis. The most fully researched regulator of MKs in this process is thrombopoietin (TPO), which is one of a complex network of hemopoietic growth factors. Generally, platelet production is regulated by TPO. The amount of TPO that is accessible to populations of hematopoietic stem and progenitor cells is regulated by the thrombopoietin receptor (c-Mpl) on megakaryocytes and platelets (*Ninos et al., 2006*). In addition to TPO, other regulators also have significant impacts on megakaryocytopoiesis. These include the interleukin cytokines family, vascular endothelial growth factors (VEGFs), SDF-1 (stromal cell-derived factor-1)/FGF-4 (fibroblast growth factor-4), insulin-like growth factor-1 (IGF-1) and other cytokines (*Zheng et al., 2008*). The majority of their effects are independent of TPO signaling to regulate MK differentiation and thrombopoiesis. Understanding the TPO-independent regulation system is crucial because it may present new options for understanding the process of development, the mechanisms behind diseases with aberrant MK and platelet formation, and the advancement of therapeutic research.

In the present study, we carried out virtual screening to predict the therapeutic activity of VLZ in the early stage and the predicted results were verified through in vitro and in vivo investigations. The molecular mechanism of the effect of VLZ on thrombocytopenia was probed through network pharmacological methods. Then experimental verification confirmed that the targeted activation of 5-HTR1A promoted the maturation and differentiation of megakaryocytes and was related to the regulation of the SRC-RAS-MEK-ERK1/2 signaling pathways. In conclusion, this study reports a combination of computational analysis and experimental validation to explore the thrombogenic effects and potential mechanisms by which 5-HTR1A drives megakaryocytopoiesis and platelet production and enriches the research on the regulatory mechanism of serotonergic signaling on hematopoietic lineages.

## Results
### Prediction of potential megakaryocytopoiesis inducers via the virtual screening model

We focused on building a drug-screening model consisting of random forests in this work. The optimal training set for predicting drug activity was selected after acquiring six training sets (*Figure 1a*) and utilizing random forest in Orange software (https://orangedatamining.com/) to perform operations on the 6 training sets. The Gini index was used to rank the importance of 20 molecular descriptors to improve predictive accuracy (*Figure 1b*). To assess the prediction capability of these training sets, we first determined the parameters of the model. When 30 trees and two attributes were assessed at each split, we found that the training set with a score of 50% provided the model with the best prediction performance, and the area under the AUC value was 0.954 (*Figure 1c*). To further validate the performance of the model, the validation set containing 2 active compounds and 15 inactive compounds was entered into the model. The findings showed that the prediction accuracy of the model was 83.33%. Subsequently, 100 chemical compounds from the Discovery Probe FDA-approved Drug Library were then predicted using the model. VLZ was projected to have a predicted score of 0.52, indicating that it possessed biological activity that promoted MK differentiation or platelet formation.

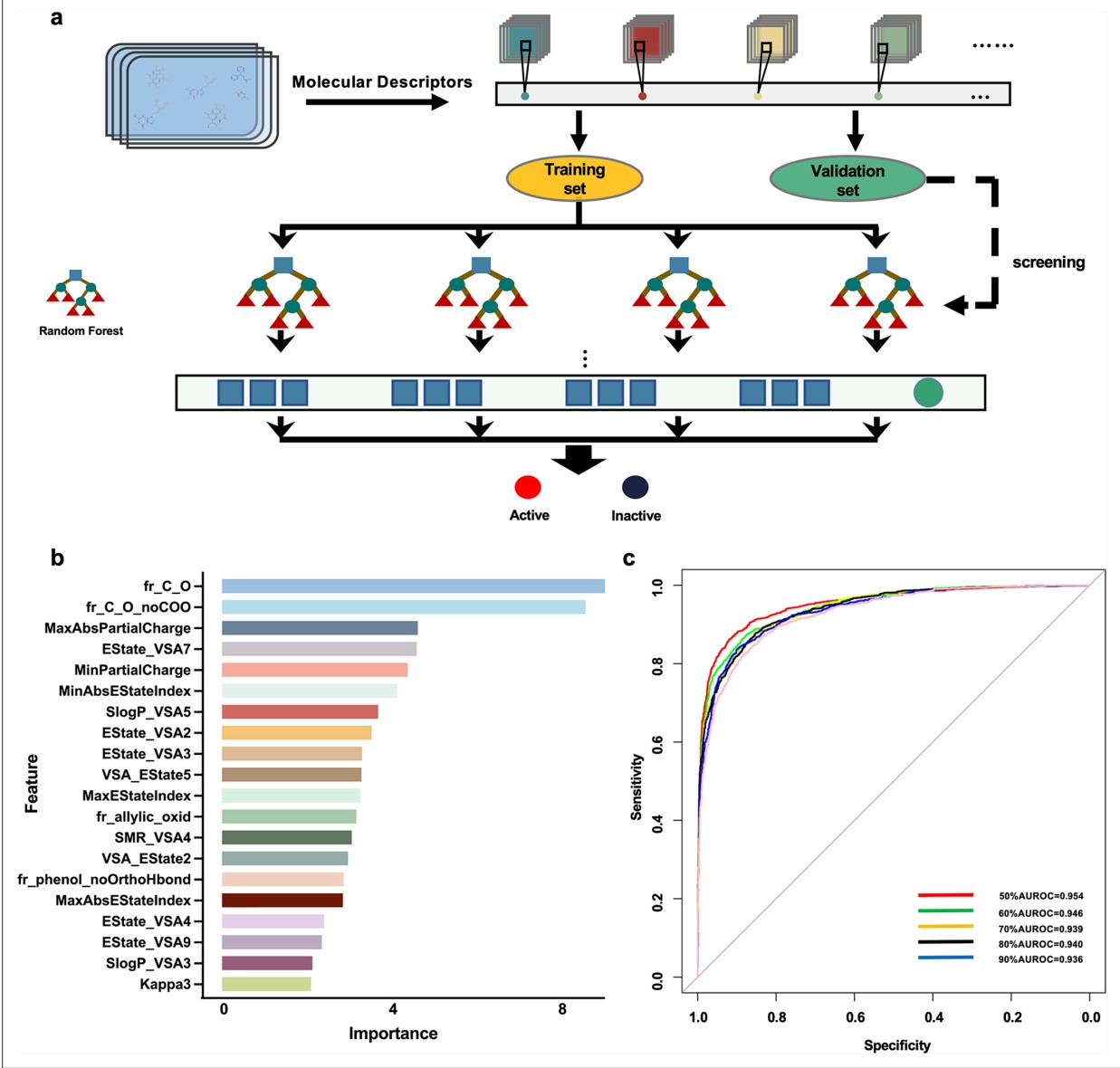

**Figure 1.** Drug screening model construction. (**a**) Flowchart for the screening model construction. (**b**) The top 20 molecular descriptors with high scores. (**c**) ROC curves of importance ratios.

## VLZ dose-dependently promotes megakaryocyte maturation and differentiation

To verify the compound activity predicted by the model, we evaluated the effect on HEL and Meg-01 cells after VLZ intervention for the specified period to search for the ideal VLZ concentration for MK differentiation. For the in vitro study, we used drug dosages of 2.5, 5, and 10 μM. LDH assay was used to determine the cytotoxicity of VLZ on HEL and Meg-01 cells. The outcomes demonstrated that 2.5, 5, and 10 μM VLZ were within a safe concentration range for MK differentiation since LDH release at days 1, 3, and 5 in the VLZ group was not different from that in the control group. (*Figure 2—figure supplement 1a,b*). Furthermore, cells started to exhibit a decrease in proliferation rates after treatment on the 3rd day, and o the 5th day, there was a concentration-dependent inhibition of proliferation (*Figure 2—figure supplement 1c,d*). This may be the result of cell differentiation. Microscopic images showed that obvious large megakaryocyte-like cells were increased in the VLZ (2.5, 5, and 10 μM) and PMA (a positive control that is known to induce MK differentiation) groups on the 5th

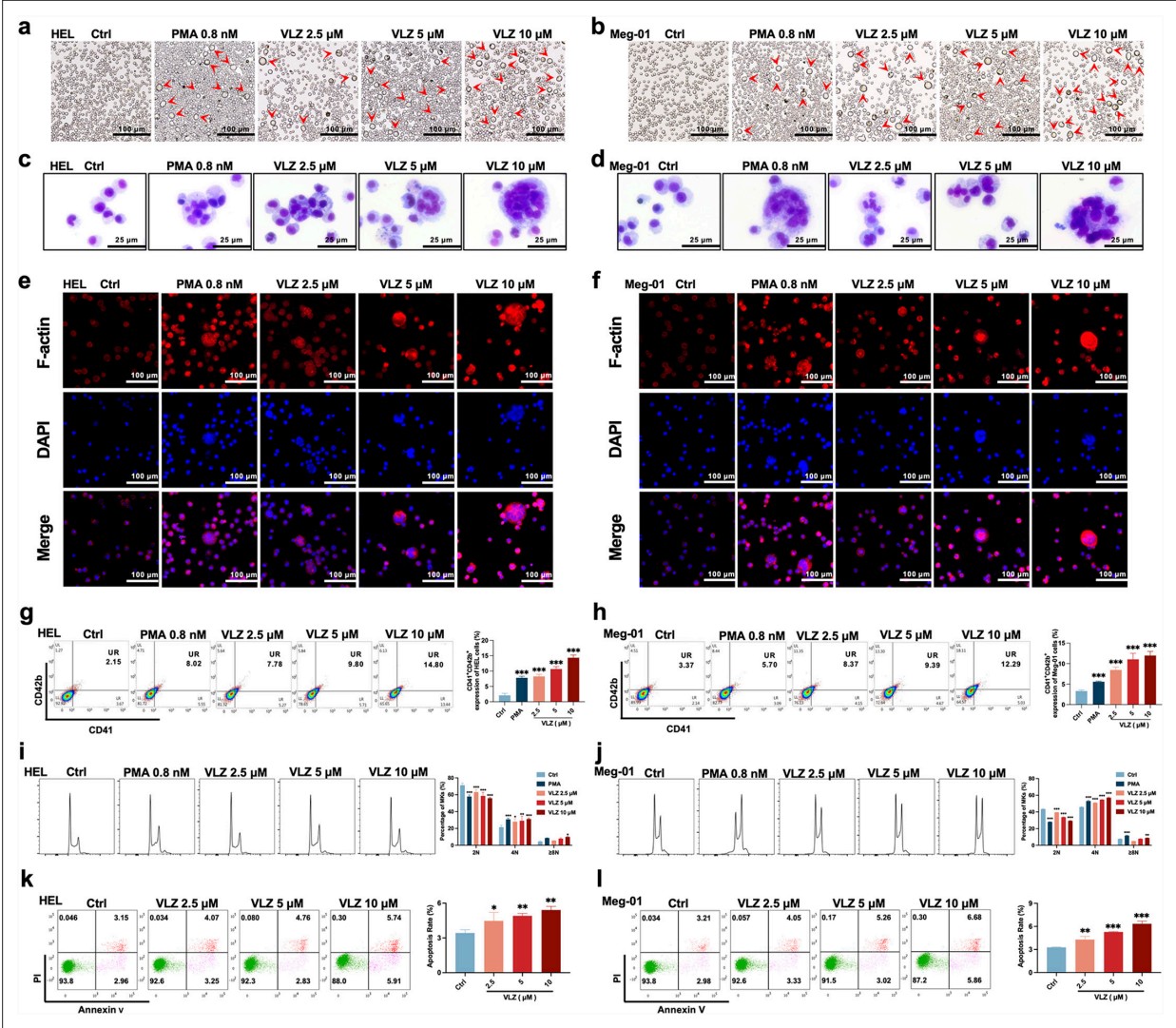

**Figure 2.** VLZ promotes megakaryocyte differentiation and enhances the DNA ploidy of HEL and Meg-01 cells. (**a, b**) Representative images of HEL and Meg-01 cells treated with different concentrations of VLZ (2.5, 5, and 10 µM) for 5 days. Bars represent 100 µm. The positive control was PMA. (**c, d**) Giemsa staining of HEL and Meg-01 cells treated with VLZ (2.5, 5, and 10 µM) or PMA for 5 days. Bars represent 25 µm. (**e, f**) Phalloidin staining of HEL and Meg-01 cells treated with VLZ (2.5, 5, and 10 µM) or PMA for 5 days. DAPI staining nuclei (blue) and Phalloidin staining of F-actin (red). Bars represent 100 µm. (**g, h**) Flow cytometry analysis of the percentage of CD41+/CD42b+ complexes surface expression on HEL and Meg-01 cells by VLZ (2.5, 5, and 10 µM) or PMA for 5 days. The histogram shows the percentage of CD41+/CD42b+ cells for each group. (**i, j**) Flow cytometry analysis of the DNA ploidy of HEL and Meg-01 cells treated with VLZ (2.5, 5, and 10 µM) or PMA for 5 days. The histogram shows the percentages of DNA ploidy. (**k, l**) Flow cytometry analysis of the cell apoptosis of HEL and Meg-01 cells treated with VLZ (2.5, 5, and 10 µM) or PMA for 5 days. Data are shown as the mean ± SD from three independent experiments. *$p \leq 0.05$, **$p \leq 0.01$, and ***$p \leq 0.001$, vs the control group.

The online version of this article includes the following figure supplement(s) for figure 2:

**Figure supplement 1.** Safe concentration of VLZ for treatment of HEL and Meg-01.

day (*Figure 2a and b*). The data suggested that VLZ is a safe inducer to inhibit MK proliferation and induce differentiation.

Given that the established role of VLZ promotes megakaryocyte generation, we continued to investigate whether VLZ exerted excellent activity in inducing MK-typical differentiation and polyploidization. Wright-Giemsa staining showed that VLZ significantly increased the nuclear-to-cytoplasm ratio, polyploid cells, and deeply stained multilobulated nuclei compared with those in the control (*Figure 2c and d*). Similarly, we evaluated the morphological alteration of VLZ-induced polyploidization, as determined by phalloidin staining. On the fifth day, significant numbers of multinucleated cells, cytoplasm enlargement, and F-actin aggregation were observed in the VLZ and PMA treatment

groups (*Figure 2e and f*). Megakaryocyte differentiation and maturation are frequently assessed using changes in the expression of distinctive molecular markers. MKs have specific antigens called CD41 and CD42b (*Jimenez et al., 2015*). After 5 days of treatment, flow cytometry analysis showed that VLZ obviously increased CD41$^+$/CD42b$^+$ cells in a dose-dependent manner in HEL and Meg-01 cells (*Figure 2g and h*). The typical feature of mature megakaryocytes is the polyploidization of the nucleus (*van Dijk et al., 2018*). Flow cytometry analysis also showed that VLZ significantly increased the population of ≥4 N DNA content in HEL and Meg-01 cells in a concentration-dependent manner after 5 days of VLZ intervention (*Figure 2i and j*). Therefore, we further determined the action of VLZ on DNA content. We know that proplatelet formation requires local apoptosis. Annexin V/PI staining flow cytometry showed that Vilazodone induced excessive apoptosis of megakaryocytes on day 5 in

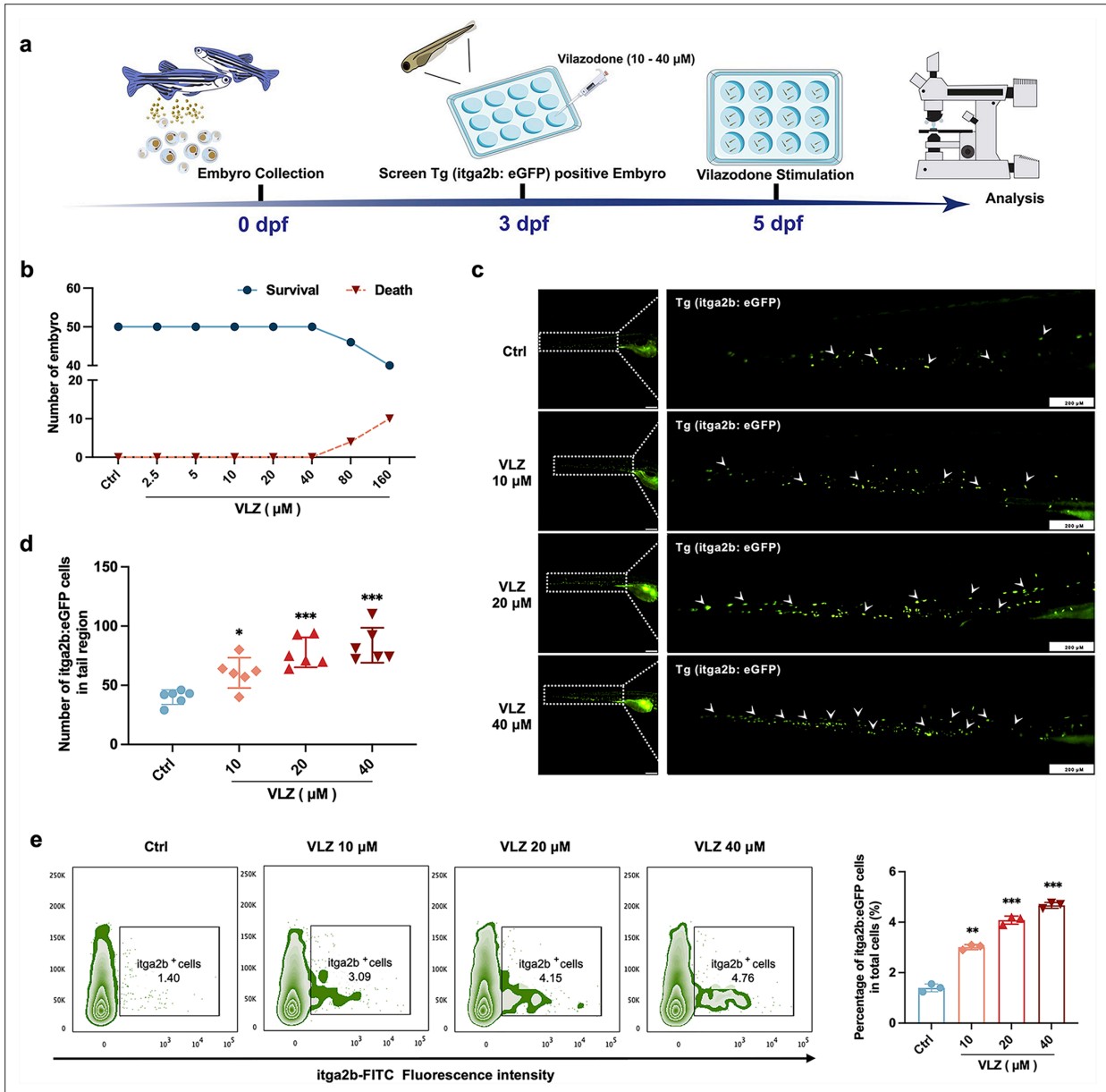

**Figure 3.** VLZ administration enhances thrombogenesis in Tg (itga2b: eGFP) transgenic zebrafish. (**a**) Timeline of the zebrafish intervention. (**b**) Toxicity effects of different doses of VLZ on fish survival and general development. (**c**) itga2b: eGFP thrombocytes in whole tail regions of control and VLZ (2.5, 5, and 10 μM) treated zebrafish. Bars represent 200 μm. (**d**) Quantification of itga2b: eGFP cells in each group (n=6 per group). (**e**) Flow cytometry analysis of GFP-labeled platelets of Tg (itga2b: eGFP) zebrafish embryos treated with different doses of VLZ. The data are shown as the mean ± SD. *p≤0.05, **p≤0.01, and ***p≤0.001, vs the control group.

a concentration-dependent manner (*Figure 2k and l*). These data suggest a dose-dependent promotion of MK maturation and differentiation by VLZ.

## VLZ increases thrombocytes in zebrafish embryos

Compared with cell models, zebrafish are an excellent model system for drug delivery and biology (*Kong et al., 2020*). We further appraised the overall effect of VLZ on promoting thrombocyte production in zebrafish (*Figure 3a*). The maximum dosage for subsequent investigations was set at 40 µM since toxicity analyses indicated that VLZ incubation had no discernible impact on the survival or general development of 3 dpf zebrafish embryos within a range of 2.5–40 µM (*Figure 3b*). The embryos of platelet labeled Tg (itga2b: eGFP) transgenic zebrafish at 3 dpf were treated with VLZ (10–40 µM), and the itga2b: eGFP thrombocytes in the tail area were carefully examined at 5 dpf. Notably, VLZ increased the quantity of GFP$^+$ cells in the tail region, which represents platelets (*Figure 3c and d*). Next, we extracted the total cells of zebrafish administered VLZ for quantification analysis of its effects on endogenous blood circulation through flow cytometry. As anticipated, there was a dose-dependent increase in the number of itga2b: eGFP cells among the total cells in the VLZ (10–40 µM) treatment groups compared to the control group, suggesting that circulating platelets were increased after VLZ stimulation (*Figure 3e*). These results demonstrate that VLZ substantially increases thrombocytes in zebrafish embryos.

## VLZ restores the quantity and function of peripheral platelets in thrombocytopenic mice

To confirm the in vivo therapeutic influence of VLZ on thrombocytopenia, a thrombocytopenia mouse model was constructed by 4 Gy X-ray total body irradiation, and KM mice were administered VLZ (2.5 mg/kg, 5 mg/kg, and 10 mg/kg) and rhTPO (recombinant human thrombopoietin, 3000 U/kg) for 12 days after radiation (*Figure 4a*). Owing to X-ray exposure, the platelet count in all irradiation groups declined to a minimum on the 7th day, whereas it remained the same in control mice. Nonetheless, VLZ was efficacious in promoting platelet production and resumption in a dose-dependent manner from the seventh day to the twelfth day, indicating that VLZ administration boosted platelet recovery when the mice encountered irradiation (*Figure 4b*). Complementary to the results obtained for zebrafish, the cell count and flow cytometry of peripheral blood showed that platelets increased significantly on the 12th day of 10 mg/kg VLZ administration in normal mice (*Figure 4—figure supplement 1*). The mean platelet volume (MPV) was likewise discovered and revealed no difference between each group, indicating that VLZ had no impact on MPV (*Figure 4c*). Additionally, we found that VLZ (5 mg/kg and 10 mg/kg) treatment improved RBC counts on the 10th day and 12th day (*Figure 4d*). However, there was no significant difference in WBCs between the TPO, VLZ, and model groups at any of the tested time points (*Figure 4—figure supplement 2*). Moreover, the expression of CD41 and CD61 in peripheral blood was determined by flow cytometry analysis. The results showed that VLZ and TPO significantly promoted the expression of CD41 and CD61, consistent with peripheral platelet counts (*Figure 4e*). Additionally, H&E staining showed no significant differences between each group in major organs, suggesting that VLZ was not systemically toxic. The visceral indexes of the liver, spleen, kidney, and thymus were examined and showed that the index of the liver was high in the model group, which recovered after VLZ or TPO administration (*Figure 4—figure supplement 3*).

To ascertain whether the platelets stimulated by VLZ therapy were functional, we subsequently studied platelet function in vivo and in vitro. P-selectin is sequestered in Weibel-Palade granules of endothelial cells and α-granules of platelets and translocated to the surface in response to various inflammatory and thrombogenic mediators (*DuRoss et al., 2021*). However, surface P-selectin (CD62P) increases after radiation damage (*Johansson et al., 2012*). Peripheral blood cells were used in a CD41/CD62P flow cytometry assay, and animals treated with VLZ and TPO had significantly reduced levels of CD41$^+$/CD62P$^+$ expression compared to the model group in the platelet resting state (*Figure 4f*). In addition, animals treated with VLZ and TPO showed dramatically increased platelet reactivity in the CD62P assay after ADP stimulation (*Figure 4g*). Moreover, platelet adhesion on solidified collagen improved after VLZ or TPO treatment (*Figure 4h*). Furthermore, we evaluated platelet aggregation induced by ADP. Compared to the model group, platelet aggregation caused by ADP was significantly strengthened in the VLZ-treated and TPO groups (*Figure 4i*). In addition, we assessed the impact of VLZ on thrombus development following vascular damage in a carotid artery thrombosis model. The

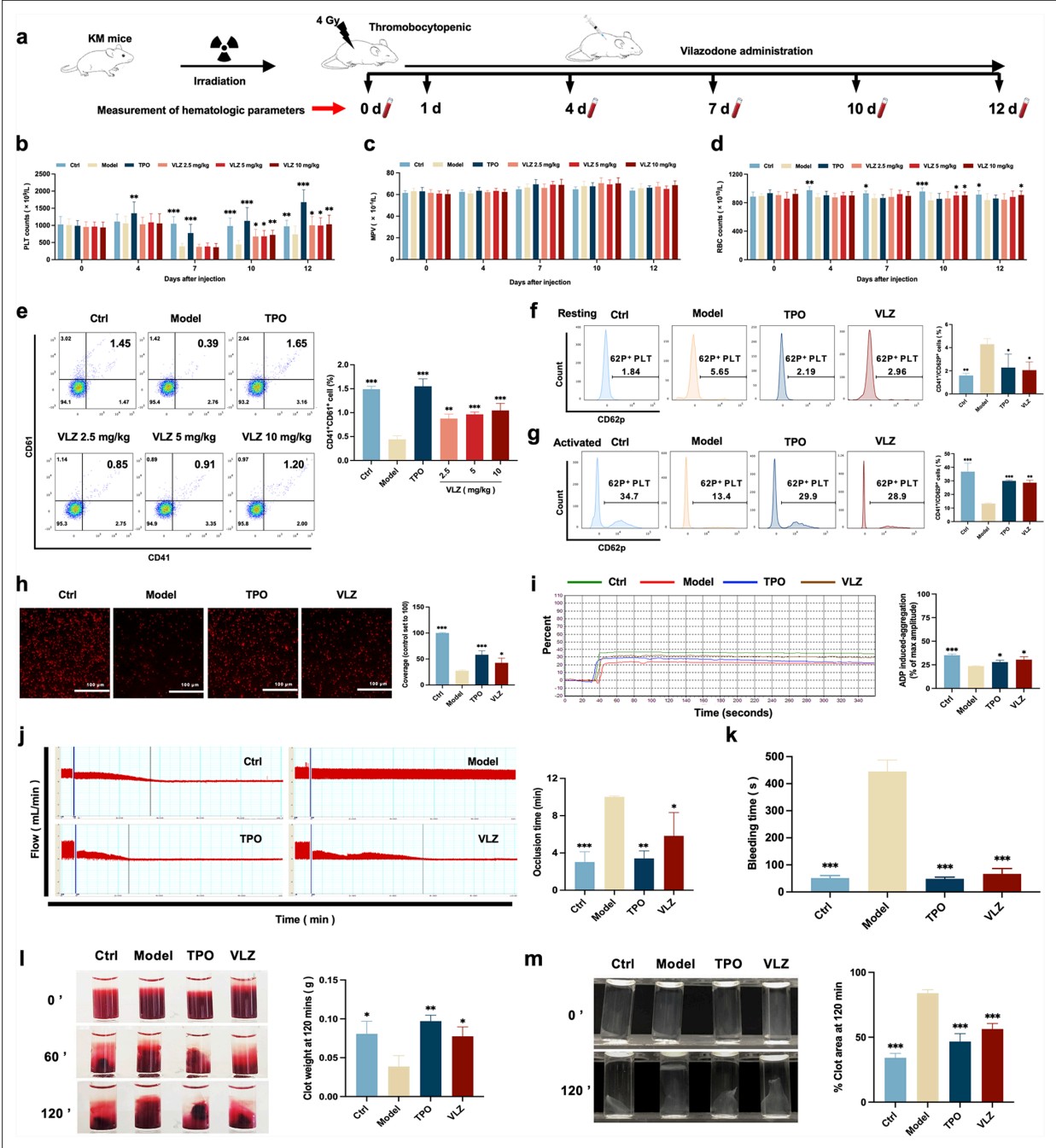

**Figure 4.** VLZ administration counteracts radiation-induced thrombocytopenia in vivo. (**a**) Schematic diagram of thrombopoietin (TPO) (3000 U/kg) and VLZ (2.5, 5, and 10 mg/kg) administration in Kunming (KM) mice with thrombocytopenia induced by IR. (**b–d**) Peripheral blood counts showing (**b**) platelet counts, (**c**) mean platelet volume (MPV) and (**d**) RBC on days 0, 4, 7, 10, and 12 post-IR. In B to E, (n=12 per group) the data are expressed as the mean ± SD, and two-way ANOVA with Tukey's multiple comparisons test was used unless otherwise specified. *p≤0.05, **p≤0.01, and ***p≤0.001, vs the model group. (**e**) Flow cytometry analysis indicates the expression of CD41 and CD61 in peripheral blood cells after receiving therapy for 12 days. The histogram represents the percentage of CD41+/CD61+ cells in each group. (**f**) Representative image of CD62p in washed platelets in each group. (**g**) Representative images of CD62p in ADP (10 mM) washed platelets in each group. (**h**) Micrographs of collagen-coated slides with the same number of platelets perfused. Red represents platelets. The histogram shows the average of red fluorescence on the whole surface by ImageJ software. (**i**) Platelet aggregation was measured by a turbidimetric aggregation-monitoring device under ADP stimulation. The histogram represents the aggregation results expressed as the maximum aggregation amplitude of platelets in each group. (**j**) The rate tracings of carotid blood flow. The histogram shows the mean carotid artery occlusion times of each group. (**k**) Tail bleeding time was measured in each group. (**l**) Representative images of whole blood clots at times 0, 60, and 120 min (end point) in each group. The histogram represents the percentage of final clot weight. (**m**) Representative images of washed

*Figure 4 continued on next page*

*Figure 4 continued*

platelet clots at time 0 and 120 min (endpoint) in each group. The histogram shows clot retraction at 120 min as a percentage of clot area from 0 min. The data represent the mean ± SD of three independent experiments. *p≤0.05, **p≤0.01, and ***p≤0.001, vs the model group.

The online version of this article includes the following figure supplement(s) for figure 4:

**Figure supplement 1.** VLZ promotes platelet increase in normal mice.

**Figure supplement 2.** WBC counts.

**Figure supplement 3.** Toxicity evaluation of VLZ in vivo.

results suggested that VLZ promoted thrombus development in mice with thrombopenia, since it took considerably less time for the VLZ-treated group than the model group to form a totally occluded thrombus after the commencement of arterial injury (*Figure 4j*). The tail bleeding time was used to quantitatively evaluate the hemostatic capacity of platelets. The findings demonstrated that mice in the VLZ and TPO treatment groups had considerably reduced tail bleeding time compared with mice in the model group (*Figure 4k*). We next examined whole blood clots and washed platelet clot development and retraction over time (120 min) following clotting triggered by thrombin and $CaCl_2$. Differences in the proportion of whole blood clot retraction were observed for VLZ and TPO and increased clot weight at the endpoint when compared with the model group (*Figure 4l*). Consistent with the results indicated for whole blood clot retraction, VLZ, and TPO completely improved washed platelet retraction compared to the model group (*Figure 4m*). Collectively, these data suggest that VLZ stimulation is conducive to platelet recovery in vivo after encountering radiation. In the meantime, it also contributed to RBC recovery. The platelet function of thrombopenia mice treated with VLZ was complete, including hemostasis and coagulation, and the response characteristics to agonists were almost normal.

## VLZ rescues bone marrow and splenic hematopoiesis after radiation injury

Bone marrow is the most important hematopoietic organ in the human body, and extramedullary hematopoiesis in the spleen may occur in response to physiological stress or illness and may serve as a substitute tissue site for bone marrow hematopoiesis (*Lucas, 2021*; *Short et al., 2019*). We thus investigated whether VLZ could promote the differentiation of hematopoietic progenitors into megakaryocytic progenitors and megakaryocytes in the BM and spleen considering the increased numbers of megakaryocytes in the VLZ-treated group. Following VLZ and TPO treatment for 12 days, H&E staining results showed a clearly increased number of megakaryocytes in the BM and spleen (*Figure 5a and b*). According to flow cytometry analysis, the percentages of c-Kit$^+$/CD41$^-$ (hematopoietic progenitors), c-Kit$^+$/CD41$^+$ (megakaryocytic progenitors), and c-Kit$^-$/CD41$^+$ (megakaryocytes) cells were considerably increased in the TPO- and VLZ-treated groups compared to the model group (*Figure 5c*). The findings imply that VLZ can stimulate megakaryocyte formation at various megakaryopoiesis stages. CD41 and CD61 were detected by flow cytometry as megakaryocyte-specific markers. The findings revealed that in the TPO- and VLZ-treated groups, the fraction of CD41$^+$/CD61$^+$ cells in the BM and spleen was noticeably higher than that in the model group, which indicated that VLZ promoted BM megakaryocyte differentiation (*Figure 5d and e*). Megakaryocytes from the BM and spleen were also examined for DNA content using flow cytometry. As anticipated, the TPO- and VLZ-treated groups greatly outnumbered the model group in increasing terms of the ploidy of BM and spleen megakaryocytes (*Figure 5f and g*). All these data indicate that VLZ promotes platelet formation in the BM and spleen of irradiated mice by rescuing megakaryopoiesis. Collectively, these results demonstrate that VLZ has excellent therapeutic effects on mice suffering from thrombocytopenia.

## Potential therapeutic targets and signaling pathway prediction based on network pharmacology and molecular docking

We used the 'drug targets' and 'disease targets' through database screening to estimate the therapeutic targets of VLZ for thrombocytopenia. After data integration, we discovered 109 targets related to VLZ treatment. A total of 4356 targets related to thrombocytopenia were obtained after duplicates were removed. Potential targets of VLZ for treating thrombocytopenia included 63 common targets (*Figure 6a*). The STRING database and Cytoscape_v3.7.1 software were used to assess the 63 putative

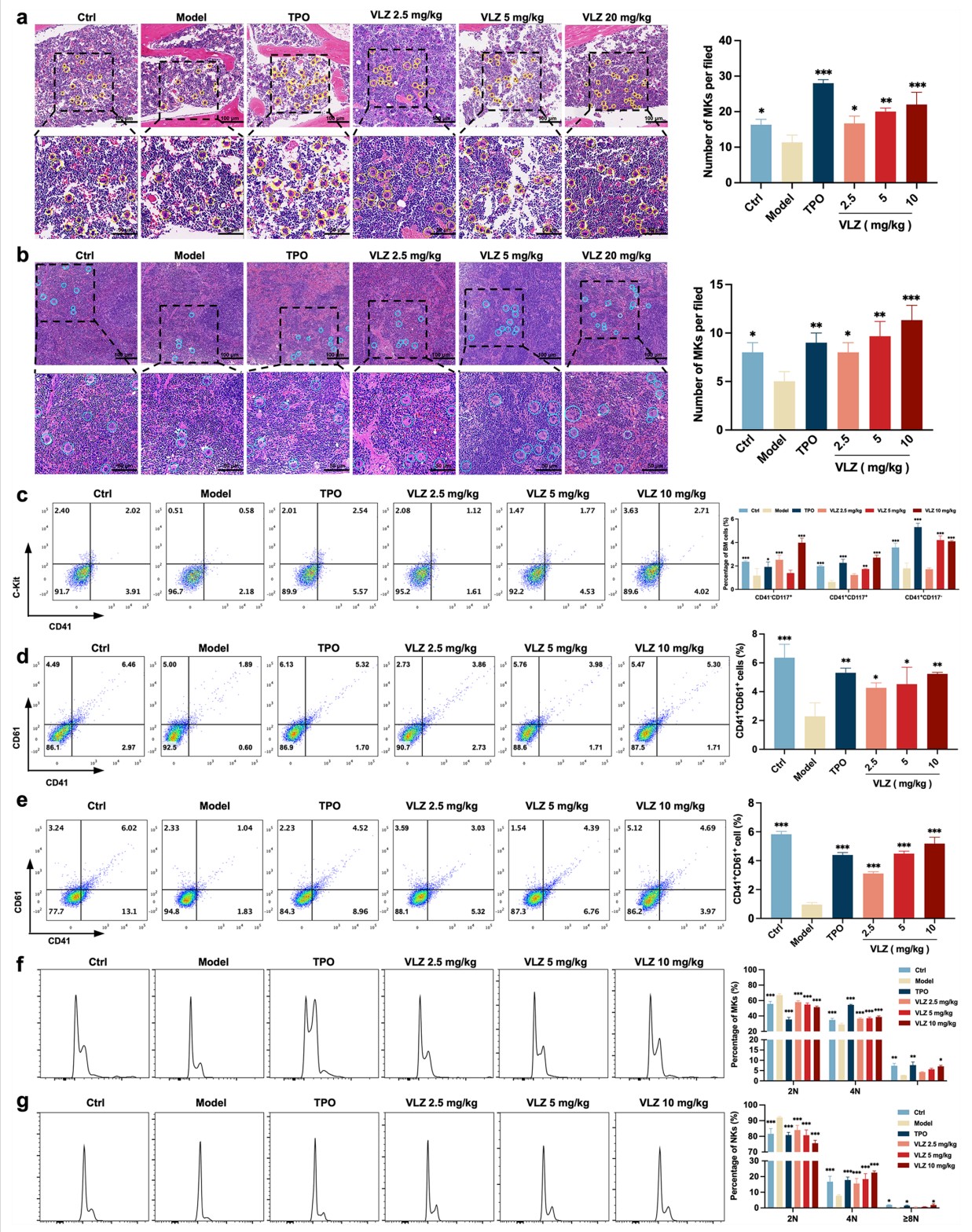

**Figure 5.** VLZ rescues bone marrow and splenic hematopoiesis after radiation injury. (**a, b**) H&E staining shows the megakaryocytes in BM and spleen after mice were treated with normal saline, thrombopoietin (TPO) (3000 U/kg), and VLZ (2.5, 5, and 10 mg/kg) for 12 days. Ten microscopy fields per sample were counted. The circles mark the megakaryocytes. The histogram shows the number of megakaryocytes in the BM and spleen in each group. (**c**) The examination of the expression of c-Kit and CD41 in each group by flow cytometry after receiving therapy for 12 days. The histogram represents the percentage of c-Kit⁺/CD41⁻, c-Kit⁺/CD41⁺, and c-Kit⁻/CD41⁺ cells in each group. (**d, e**) Flow cytometry analysis indicates the expression of CD41

*Figure 5 continued on next page*

Figure 5 continued

and CD61 in BM and spleen cells after receiving therapy for 12 days. The histogram represents the percentage of CD41⁺/CD61⁺ cells in each group. (**f, g**) Flow cytometry analysis indicates the cell ploidy of BM and spleen cells of each group. The histogram represents the cell ploidy in the BM and spleen cells of each group. All data represent the mean ± SD of three independent experiments. *p≤0.05, **p≤0.01, and ***p≤0.001, vs the model group.

therapeutic targets and create an interaction network (**Figure 6b**). Then, we performed a topology analysis to identify the core targets. The top 6 target proteins of MAPK1, HTR1A, SRC, NFE2, ABL1, and MCL1 were regarded as core proteins (**Figure 6c**). Utilizing molecular docking stimulation, the direct connection between VLZ and the primary targets was further investigated. Molecular affinity for the target was indicated by a docking score of 4.52 kcal mol⁻¹, with a docking score >5 kcal mol⁻¹ being considered indicative of a strong binding affinity (**Hsin et al., 2013**). According to the docking results, the binding scores of MAPK1 and SRC with VLZ were 8.97 and 5.31, respectively, which indicated a strong affinity for VLZ (**Figure 6d**). GO and KEGG enrichment analyses were conducted to elucidate the potential underlying mechanism of VLZ action on thrombocytopenia. The outcomes demonstrated that VLZ treatment for thrombocytopenia was associated with a total of 58 GO terms (**Figure 6e**). These GO annotations mainly involve positive regulation of MAP kinase activity, positive regulation of ERK1 and ERK2 cascade, stem cell differentiation and positive regulation of smooth muscle cell migration, growth factor binding, transcription factor binding, and ATP binding. In summary, the top 10 CC, BP, and the most VLZ-related process-enriched MF terms were sorted independently in descending order of p values. (**Figure 6f**). KEGG enrichment analysis revealed that VLZ regulatory action in VLZ was significantly enriched in the MAPK signaling pathway, RAS signaling pathway, PI3K-Akt signaling pathway, and Rap1 signaling pathway. The 20 most enriched pathways were arranged in descending order of p values (**Figure 6g**). Therefore, these results suggest that VLZ might protect against thrombocytopenia via activation of the SRC and MAPK (RAS/MEK/ERK) signaling pathways.

## The SRC/MAPK signaling pathway is activated during VLZ-facilitated megakaryopoiesis

Given the emphasis of the in-silico prediction results emphasized the involvement of the MAPK signaling pathway and RAS signaling pathway in a latent regulatory mechanism of VLZ, we proceeded to investigate the expression of the core proteins within this pathway, which are diverse and embody VLZ-prompted pathway regulation: (p-)SRC, RAS, (p-)MEK, and (p-)ERK1/2 (**Figure 7a–d**). After 5 days, the phosphorylation levels of SRC, MEK, ERK1/2 and RAS were significantly changed in VLZ (2.5, 5, and 10 μM)-treated Meg-01 cells. To produce proplatelets and differentiate and mature megakaryocytes, the transcription factors NF-E2, EGR1, FOS, and RUNX1 link upstream molecular signals with downstream transcript expression (**Noetzli et al., 2019**; **Kim et al., 2019**). After VLZ treatment at 2.5 μM, 5 μM, and 10 μM, we further confirmed that VLZ treatment led to a dose-dependent increase in the expression of NF-E2, EGR1, FOS, and RUNX1 at the translational level (**Figure 7e–h**). Additionally, immunofluorescence research verified that VLZ boosted the expression of NF-E2 and RUNX1 in Meg-01 cells (**Figure 7i and j**). Finally, we used a blocking technique to determine whether they controlled the megakaryocyte differentiation that VLZ generated. The ERK1/2-specific inhibitor SCH772984 was used to block the phosphorylation of ERK1/2. Flow cytometry analysis demonstrated that VLZ-induced acceleration of megakaryocyte differentiation was interrupted by SCH772984 (3 μM) therapy (**Figure 7k**). This result supports the molecular mechanism by which VLZ activates the SRC/RAS/MEK/ERK1/2 cascade to promote the mRNA expression and nuclear translocation of related transcription factors to induce MK maturation and platelet production.

## VLZ stimulated MK polyploidization and maturation via activation of the 5-HT1A receptor

Despite VLZ being an agonist of 5-HTR1A, to further validate whether the target also acted on MKs, we explored whether VLZ interacts specifically with 5-HTR1A to promote MK differentiation. As a preliminary study, the expression of 5-HTR1A in Meg-01 cells was evaluated. We also demonstrated that VLZ doses increased the expression of 5-HTR1A (**Figure 8a and d**). A DARTS assay was then employed to observe the binding performance of VLZ to 5-HTR1A on Meg-01 cells. The DARTS results demonstrated that VLZ administration increased the stability of 5-HTR1A against protease-induced

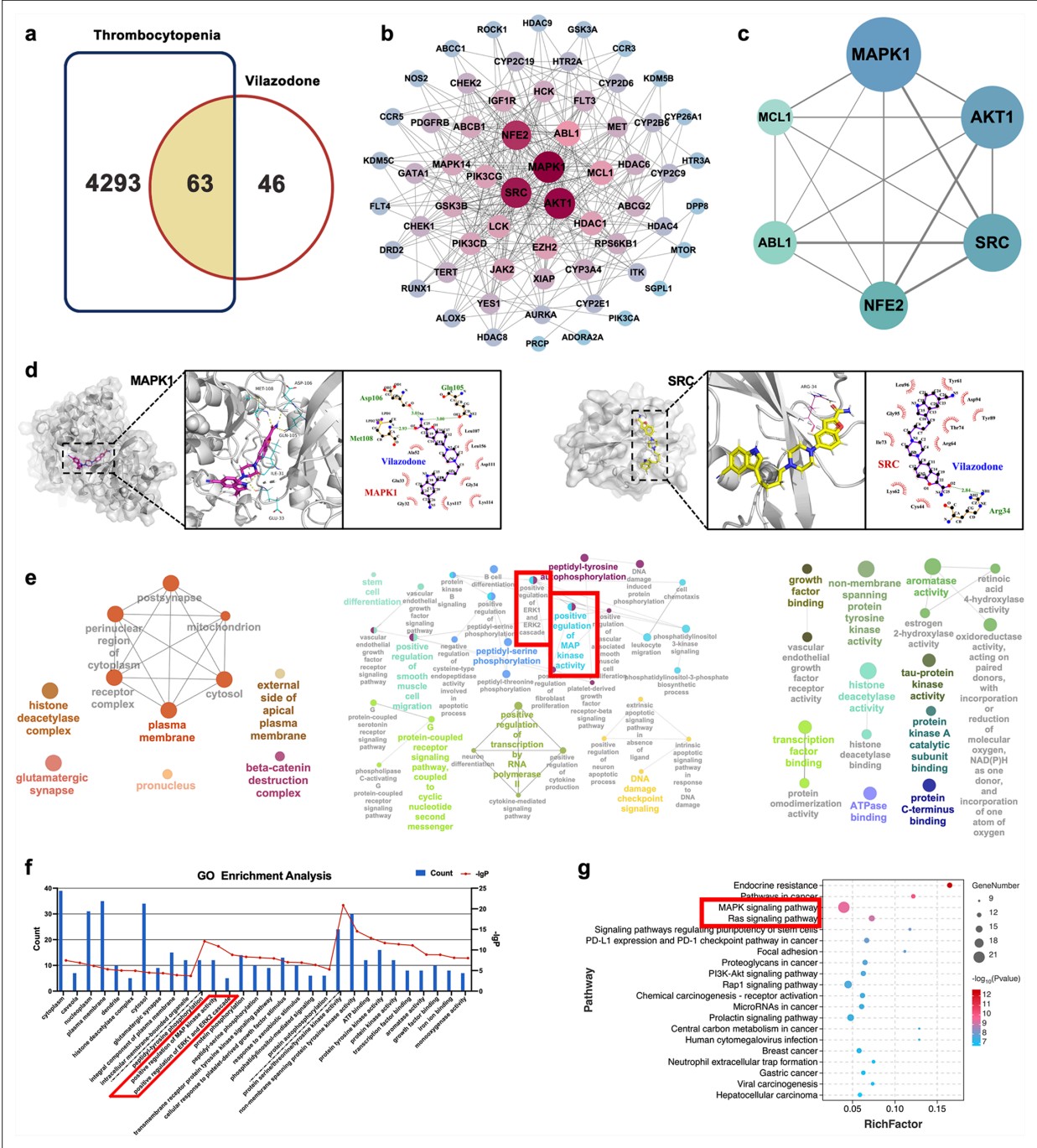

**Figure 6.** Network pharmacology and molecular docking analysis of VLZ activity in thrombocytopenia. (**a**) Venn diagram of the common targets of VLZ and thrombocytopenia. (**b**) The VLZ-targets-Thrombocytopenia network that was constructed by Cytoscape_v3.7.1 software. (**c**) Protein-protein interaction (PPI) network based on the core targets of VLZ against thrombocytopenia through the screening conditions of Degree >12, BC >0.01105525, CC >0.52777778. (**d**) Molecular docking shows the binding ability between VLZ and its core targets (MAPK1 and SRC). (**e**) Visualization of cellular component, biological process, and molecular function enrichment analysis. (**f**) The top 10 of CC, BP, and MF terms showing the greatest enrichment with VLZ-related processes are listed in ascending order of *P* values.(**g**) The top 20 of KEGG pathways enrichment analysis for the mechanisms of VLZ against thrombocytopenia. A higher richness factor indicates more enrichment. The number of genes enriched in each pathway is represented by the size of the bubble. The p value range is shown by the color of the bubble.

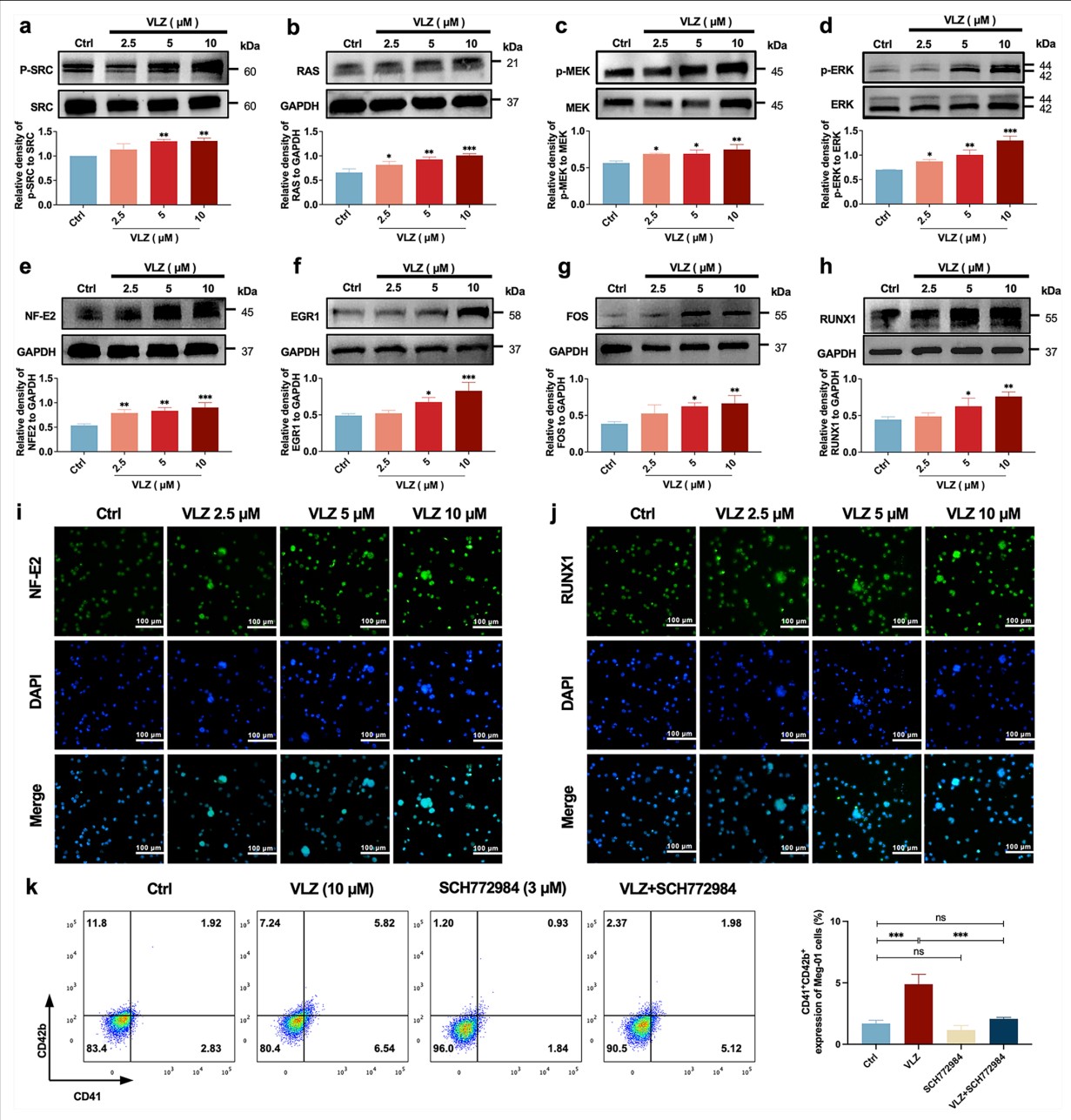

**Figure 7.** VLZ induces megakaryocyte (MK) maturation and differentiation by activating the SRC/RAS/MEK/ERK signaling pathway. (**a-h**) The expression of SRC, RAS, MEK, ERK, NF-E2, EGR1, FOS, and RUNX1 were detected by western blot after Meg-01 cells were treated with VLZ (2.5, 5, and 10 μM) for 5 days (n=3 per group). (**i, j**) Immunofluorescence analysis of the expression of NF-E2 and RUNX1 in Meg-01 cells after VLZ (2.5, 5, and 10 μM) intervention for 5 days. Cells were stained with DAPI for nuclei (blue) and antibodies for NF-E2 (green). Bars represent 100 μm. (**k**) Flow cytometry analysis of the percentage of CD41+ CD42b+ complexes surface expression on HEL and Meg-01 cells treated with VLZ (10 μM), SCH772984 (3 μM), and VLZ (10 μM)+SCH772984 (3 μM) treated for 5 days. The histogram shows the percentage of CD41+/CD42b+ cells for each group. The data represent the mean ± SD of three independent experiments. *p≤0.05, **p≤0.01, and ***p≤0.001, ns: no significance, vs the control group.

The online version of this article includes the following source data for figure 7:

**Source data 1.** Original file for the Western blot analysis in *Figure 7*.

**Source data 2.** PDF containing *Figure 7* and original scans of the relevant Western blot analysis with highlighted bands and sample labels.

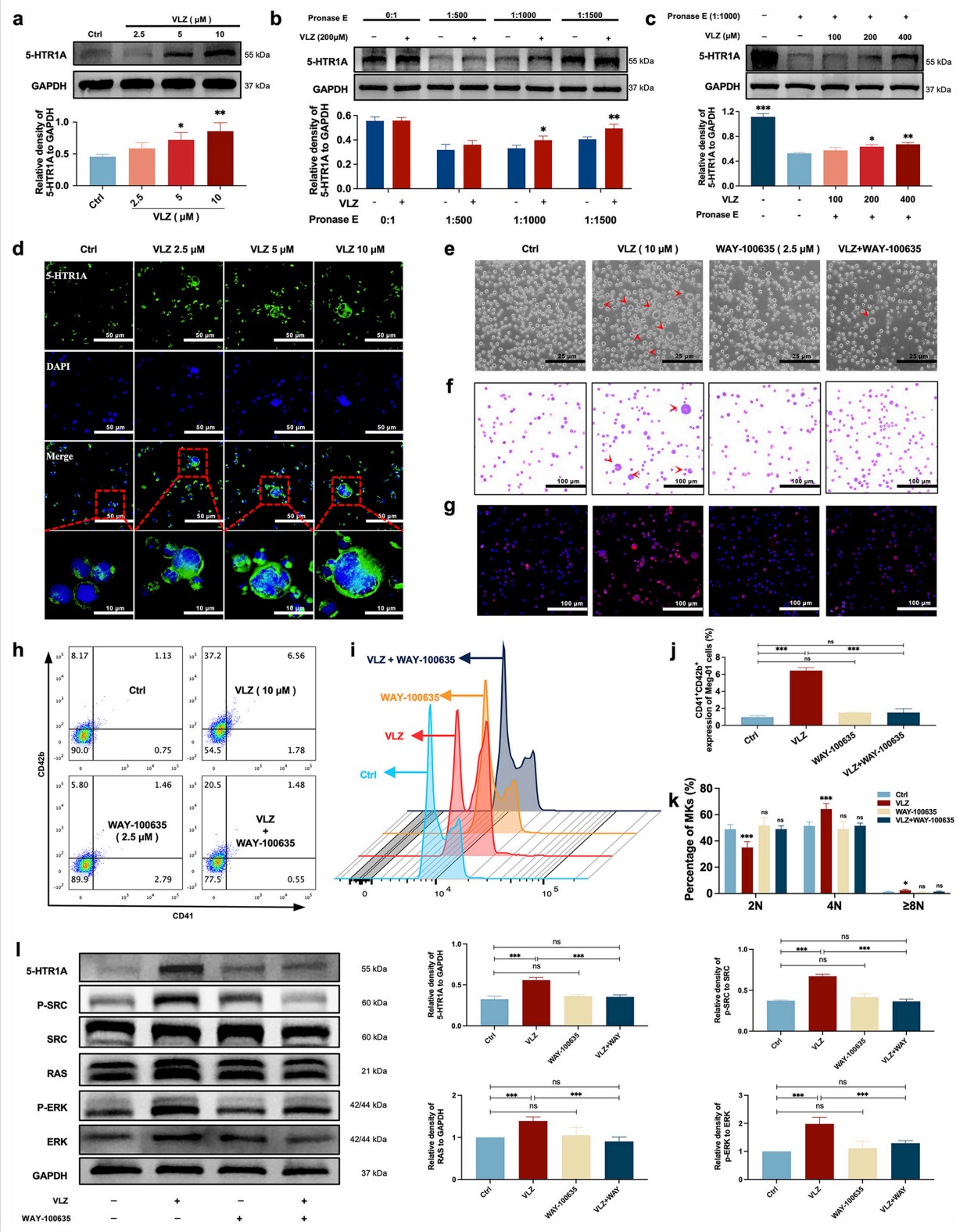

**Figure 8.** VLZ stimulates megakaryocytopoiesis via the 5-HT1A receptor. (**a**) Representative immunoblot images and biochemical quantification of 5-HTR1A after treatment with VLZ (2.5, 5, and 10 µM) in Meg-01 cells for 5 days (n=3 per group). (**b**) The DARTS assay for target validation. 5-HTR1A protein stability was increased upon VLZ (200 µM) treatment in Meg-01 lysates. Pronase was added using several dilutions (1:500, 1:1000, or 1500) from 50 µg/mL stock for 10 min at 40 °C (n=3 per group). (**c**) The DARTS assay demonstrated the dose-dependent binding of VLZ to 5-HTR1A in Meg-01 cells. Treatment with pronase (1:1000) was conducted for 10 min at 40 °C (n=3 per group). (**d**) Immunofluorescence analysis of the expression of

*Figure 8 continued on next page*

*Figure 8 continued*

5-HTR1A in Meg-01 cells after VLZ (2.5, 5, and 10 µM) intervention for 5 days. Cells were stained with DAPI for nuclei (blue) and antibodies for 5-HTR1A (green). Bars represent 100 µm. (**e–k**) Meg-01 cells were treated with VLZ (10 µM), WAY-100635 (2.5 µM), VLZ (10 µM)+WAY-100635 (2.5 µM) for 5 days. (**e**) Representative images, bars represent 25 µm. (**f**) Giemsa staining of Meg-01 cells, bars represent 100 µm. (**g**) Phalloidin staining of Meg-01 cells, bars represent 100 µm. (**h, i**) Flow cytometry analysis of the expression of CD41/CD42b and the DNA ploidy. (**j, k**) The histogram shows the percentage of CD41+/CD42b+ cells and DNA ploidy for each group. (**l**) Western blot analysis of 5-HTR1A, RAS and ERK expression after Meg-01 cells were treated with VLZ (10 µM), WAY-100635 (2.5 µM), and VLZ (10 µM)+WAY-100635 (2.5 µM) for 5 days. The histogram shows the expression of 5-HTR1A, RAS, and ERK in each group (n=3 per group). The data represent the mean ± SD of three independent experiments. *p≤0.05, **p≤0.01, and ***p≤0.001, ns: no significance, vs the control group.

The online version of this article includes the following source data for figure 8:

**Source data 1.** Original file for the Western blot analysis in *Figure 8*.

**Source data 2.** PDF containing *Figure 8* and original scans of the relevant Western blot analysis with highlighted bands and sample labels.

degradation in a concentration-dependent manner (*Figure 8b and c*). To obtain further evidence for the targeting of VLZ to 5-HTR1A, WAY-100635 (2.5 µM, 5-HT1A receptor antagonist) was applied with VLZ. In this case, the observed effects in vitro were mediated via a 5-HTR1A-dependent mechanism, as WAY-100635 administration blocked the accelerative effects of VLZ in Meg-01 cells (*Figure 8e–k*). Once activation of 5-HTR1A induced by VLZ was blocked, the expression of RAS and phosphorylation of SRC and ERK1/2 were significantly inhibited (*Figure 8l*). We indicated that the agonistic action of 5-HTR1A induced by VLZ was responsible for the activation of RAS and phosphorylation of ERK1/2. Therefore, we further demonstrated that VLZ activates 5-HTR1A on MKs to regulate the SRC/RAS/MEK/ERK signaling pathway, encouraging the nuclear translocation and mRNA expression of relevant transcription factors to trigger MK maturation and platelet formation.

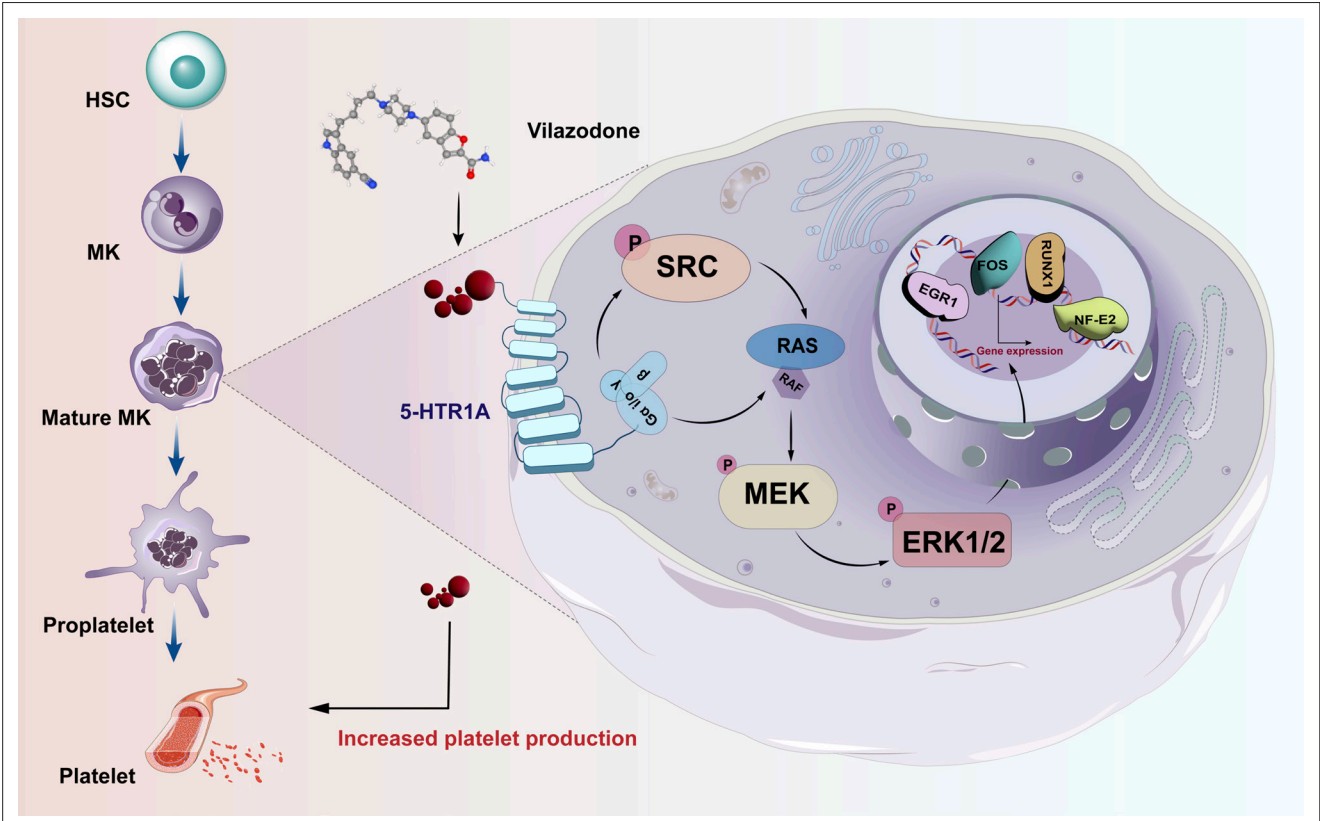

**Figure 9.** Schematic representation of the mechanism by which VLZ regulates megakaryocyte differentiation and thrombopoiesis.

## Discussion

Thrombocytopenia caused by long-term radiotherapy and chemotherapy exists in cancer treatment (*Tkaczynski et al., 2018*; *Kiang et al., 2014*). This complication seriously affects the prognosis and endangers the life of patients (*Mouthon et al., 1999*). Thrombocytopenia remains a challenging disease despite multiple advances in drug research radiotherapies, and chemotherapies Currently, there are not entirely reliable and secure medications for the rapid treatment of thrombocytopenia. Although TPO-RA is a classical clinical treatment, it is also ineffective in individuals with TPO resistance or c-Mpl gene loss-of-function mutations (*Ghanima et al., 2019*; *Poston and Gernsheimer, 2019*; *van den Oudenrijn et al., 2000*). In addition, there are few agonists targeting TPOR, so the research and development of novel non-TPOR-dependent targets and agonists is currently a key strategy for the treatment of thrombocytopenia (*Jenkins et al., 2007*; *Nakamura et al., 2006*). Several studies have demonstrated the mechanisms underlying 5-HT-regulated hematopoietic lineages. It is crucial and necessary to investigate this TPO-independent signaling pathway. Here, we identified 5-HTR1A as an attractive target, VLZ could directly bind to 5-HTR1A to enhance megakaryocyte differentiation and platelet formation by facilitating the activation of the SRC and MAPK signaling pathways at the same time (*Figure 9*).

ML has unique advantages in compound activity prediction and screening (*Bannigan et al., 2021*). In this study, we relied on a random forest-based drug screening model to characterize the hematopoietic activity of VLZ. Excitingly, VLZ demonstrated an activity prediction score worthy of further investigation, and its excellent hematopoietic activity was validated in subsequent experiments. In a safe therapeutic range, it promoted MK differentiation and maturation in a concentration-dependent manner in vitro. Furthermore, because of its advantages of in vivo complexity and in vitro high-throughput screening simplicity, zebrafish is a commonly recognized vertebrate model for research in many fields, including developmental biology, toxicology, and drug discovery (*Kong et al., 2020*). Previous studies showed the construction and application of megakaryocyte production in Tg (itga2b: eGFP) zebrafish (*Wang et al., 2023*). At the same time, VLZ significantly promoted the hematopoietic system of transgenic zebrafish and increased platelet-labeled fluorescence. To verify its therapeutic effect on thrombocytopenia, we constructed a mouse model of thrombocytopenia by whole-body radiation. After administration of different doses of VLZ for 12 days, its concentration-dependently increased platelet counts from day 10 onward, shortening the time to normalization of platelets. Under normal physiological conditions, it increased platelets in nonirradiated mice in a manner similar to the zebrafish model. In addition, we found that VLZ also had a significant effect on the recovery of red blood cells. In addition, to determine whether VLZ restores platelet function in radiation injury and whether its inhibitory effect on platelet SERT affects generated platelets, we explored platelet function in vitro and in vivo. We found that VLZ restored hemostasis thrombosis and platelet function in radiation model mice. As previously mentioned, SSRI drugs can inhibit the platelet transporter SERT and affect its function (*Scharf, 2012*). However, VLZ appeared to produce SERT-independent enhancement of platelet function in irradiated mice. Of course, it is possible that the doses and periods of our experiments did not have a significant influence on platelet inhibition. To determine which hematopoietic tissues are responsible for the findings, we subsequently analyzed the bone marrow (BM) and spleen. The results from H&E staining and flow cytometry demonstrated that VLZ therapy stimulated the differentiation of BM hematopoietic progenitor cells and megakaryocytes, thereby producing and activating platelets. Similarly, VLZ increased splenic MK numbers, promoting extra splenial hematopoiesis. These data suggest that VLZ treats thrombocytopenia by enhancing bone marrow and splenic megakaryopoiesis and thrombosis. Simultaneously, VLZ compensated for the extra slight inhibition by promoting the production of new platelets in vivo. Admittedly, the overall effect of VLZ is to gradually restore the functional system of platelets damaged by radiation to normal. This suggests that VLZ may enhance platelet function in thrombocytopenic mice independently of SERT-mediated 5-HT transport, which needs to be further investigated.

Progenitor cells in the bone marrow undergo a process known as megakaryopoiesis in which they grow into megakaryocytes, which then create the platelets required for healthy hemostasis (*Crispino, 2022*). Network pharmacology and molecular docking analysis predicted that VLZ may regulate megakaryocyte differentiation through the SRC and MAPK signaling pathways. Tyrosine-protein kinase proto-oncogene SRC, including other SRC family kinases (SFK), is crucial for controlling signal transduction by various cell surface receptors in response to alterations in the cellular environment

(*De Kock and Freson, 2020*). SRC tyrosine kinases are signaling mediators that occur before MAPK activation and gene expression that is induced by cytokines (*Singer et al., 2011*). Serine/threonine kinases known as mitogen-activated protein kinases (MAPKs) regulate important cellular responses in eukaryotic organisms and play a role in proliferation, migration, differentiation, and death (*Roux and Blenis, 2004*). Extracellular signal-related kinases (ERKs) (also known as p44/p42, ERK1/2), ERK5, p38MAPKs, and c-Jun amino-terminal kinases are the four primary protein subgroups that make up the MAPK family (*Anderson et al., 1990*). Further analysis of GO and KEGG enrichment led us to target the RAS-MEK/ERK signaling pathway. These intracellular signaling pathways play a pivotal role in many essential cellular processes, including proliferation and differentiation. Likewise, by additional validation, we showed that VLZ can activate SRC, RAS, MEK, and ERK and that it may have an impact on the signaling pathways linked to differentiation. Additionally, cytokines tightly control MK differentiation and thrombosis. Studies have revealed that NF-E2 is necessary for platelet release, proplatelet production, and megakaryocyte differentiation and maturation (*Machlus et al., 2016*; *Chen et al., 2007*). EGR1 is an early growth response transcription factor (TF) that regulates a wide array of genes and plays a major role in MKs, including cell proliferation, differentiation, and apoptosis (*Kanaji et al., 2018*; *Crist et al., 2013*). Several megakaryocytic cell surface indicators and other auxiliary genes require FOS for expression (*Limb et al., 2009*; *Ahmad et al., 2017*). The transcription factor RUNX1 (previously called AML-1 and CBFα2) is an essential regulator of definitive hematopoietic stem cell (HSC) ontogeny, HSC self-renewal, MK maturation, and lymphocyte differentiation (*Huang et al., 2012*; *Raghuwanshi et al., 2020*). The results demonstrated that VLZ activated these TFs in a concentration-dependent manner. To further investigate whether the MAPK signaling pathway was responsible for VLZ-induced MK differentiation, we blocked this signaling pathway with its inhibitor SCH772984. As anticipated, we discovered that blocking the MEK/ERK signaling pathway eliminated the ability of VLZ to cause MK differentiation. Thus far, the signaling pathway predicted by network pharmacology has been verified. VLZ induces the maturation and differentiation of MKs by promoting the expression of related transcription factors by regulating the SRC/MAPK signaling pathway.

As previously mentioned, VLZ is a 5-HT1A receptor agonist (*Wang et al., 2015*). Previous studies have shown that the 5-HT tryptamine receptor signalingergic system is involved in the process from embryonic to stem progenitor cells and subsequent differentiation. However, most research has discovered and explored the effects of 5-HT2R on the hematopoietic system. The 5-HT receptor family is very large, and the interreceptor interaction and signal networks are complex. The regulation of different systems is both similar and different. To determine whether 5-HTR1A is involved in the induction of MK differentiation, we firstly validated the expression of 5-HTR1A on MKs. We first found that 5-HTR1A was expressed on Meg-01 cells and that VLZ could directly bind to the receptor. To further determine whether 5-HTR1A activates the MAPK signaling pathway and thus induces MK differentiation, we blocked the receptor with its antagonist WAY-100635. As expected, we found that 5-HTR1A inhibition completely abolished the induction of MK differentiation by VLZ and was accompanied by a decrease in the expression of the downstream signaling pathways RAS and ERK. Collectively, the data suggest that 5-HTR1A mediates the MK differentiation and maturation of the VLZ and that this effect is accomplished by 5-HTR1A through regulation of the MAPK (RAS/ERK) signaling pathway. In conclusion, we showed that VLZ stimulates MK differentiation and platelet formation through the 5-HTR1A/SRC/MAPK pathway. Our findings highlight the importance of 5-HT receptor signaling for promoting hematopoiesis and provide new insights into the underlying mechanism of VLZ.

Taken together, our study elucidated the effects and mechanisms of VLZ in the treatment of thrombocytopenia through a combination of computational analysis and experimental validation. We revealed a previously unknown interplay in which the expression of 5-HTR1A on MKs and the regulation of the SRC/RAS/MEK/ERK signaling pathway are involved in the effects of VLZ. Our study shows the therapeutic effects and potential mechanisms of VLZ in thrombocytopenia and enriches the study of the mechanisms of 5-HT receptor signalingergic regulation of the hematopoietic spectrum.

## Materials and methods
### Construction of a virtual screening model based on random forest
The virtual screening flowchart is depicted in *Figure 1a*. First, we established a database of 39 active small molecule compounds that could promote MK differentiation or platelet generation and

691 inactive compounds. RDKit software (*http://www.rdkit.org*) was used to calculate the molecular descriptors of the 730 compounds to obtain their corresponding values. Molecular Descriptors contain multiple molecular properties, including MolWt, LabuteASA, Chi2n, and TPSA. The values of molecular descriptors such as fr_thiocyan, fr_amidine, NumRadicalElectrons and SMR_VSA8 are meaningless. All of them need to be deleted to avoid affecting the results. A total of 160 Molecular Descriptors and their corresponding values are residual. The training sets consisted of data obtained from the remaining 713 small molecule compounds, while the validation sets comprised a random selection of 2 active and 15 inactive small molecule compounds. The equilibrium between the positive and negative samples was achieved using the synthetic minority oversampling technique (SMOTE) (*Mathew et al., 2018*). According to the value of the Gini index, the data of the top 50%, 60%, 70%, 80%, 90%, and 100% of the fraction were obtained, and six training sets were generated. Then, 6 training sets were entered into the random forest. The area under the ROC curve (AUC) value was used to choose the greatest training set for the random forest, and the validation set was utilized to test the model. Eventually, the activities of compounds were predicted.

## Reagents and antibodies

Vilazodone hydrochloride (V823988, purity of 98.0%) was obtained from Macklin (Shanghai, China). The substance was dissolved in DMSO and then stored at a concentration of 10 mM/L. Antibodies against p-SRC (6943 S), SRC (2109 S), p-MEK (2338 S), MEK1/2 (4694S), p-ERK1/2 (4370S), ERK1/2 (4696S) were obtained from Cell Signaling Technology (Boston, MA, USA), against NFE2 (11089–1-AP), RUNX1 (25315–1-AP), FOS (66590–1-IG), GAPDH (60004–1-IG) were obtained from Proteintech (Wuhan, China), against RAS (T56672S), EGR1 (T57177) was obtained from Abmart (Shanghai, China), against HTR1A (BA1391) was obtained from Boster Bio-Engineering Co. (Wuhan, China). The second antibody against rabbit (4414) and against mouse (4410) were obtained from Cell Signaling Technology (Boston, MA, USA).

## Cell culture

Human erythroleukemia cell line HEL and human megakaryoblastic cell line Meg-01, mycoplasma tested, were purchased from the American Type Culture Collection (ATCC) (Manassas, VA, USA) and used throughout the experiments. Cells were kept at 37 °C in a humid incubator with 5% CO2 and grown in RPMI Medium 1640 Basic (CAT: SP032020500, Sperikon Life Science & Biotechnology Co. Ltd., Chengdu, China) with 10% fetal bovine serum (FBS, Gibco, Thermo Fisher Scientific, Waltham, MA, USA), 100 U/mL penicillin and 100 µg/mL streptomycin (Beyotime, Chengdu, China).

## Cell proliferation assay

HEL and Meg-01 cells were treated with VLZ in 96-well flat-bottomed plates for 1–5 days. The amount of lactic acid dehydrogenase (LDH) released from the cells was measured to evaluate cytotoxicity. We used an LDH cytotoxicity assay kit (Beyotime, Jiangsu, China). The maximum LDH control was used to calculate the total amount of LDH present in the cells. The absorbance was measured at 490 nm. A Cell Counting Kit-8 (CCK-8) assay (Dojindo, Kumamoto, Japan) was used to assess the proliferation of HEL and Meg-01 cells. Briefly, $5 \times 10^3$ cells were seeded in 96-well plates and then exposed to different concentrations of VLZ (2.5, 5, and 10 µM) at 37 °C and 5% $CO_2$ for 5 days. Each well received CCK-8 solution after the treatment, which was then incubated for 2 hr at 37 °C. The absorbance (OD) was measured at 450 nm.

## Analysis of cell morphology

HEL and Meg-01 cells ($4 \times 10^4$ cells/well) were seeded in 6-well plates. Microscopy (Olympus CKX53, Japan) at 10×resolution was used to detect the change in cell shape after treatment with VLZ (2.5, 5, and 10 µM) or the positive control PMA (0.8 nM) for 5 days. Giemsa staining was used to analyze the formation of giant multinucleated cells. Cells were spread out on slides using a cytocentrifuge and centrifuged at 200 rpm while being treated with a fixing solution (methanol: glacial acetic acid = 3:1 (v/v)). Giemsa solution (Solarbio, Beijing, China) was added to each sample for 15 min. Finally, the stained cells were photographed under an electron microscope. Changes in F-actin were analyzed by phalloidin staining. Cells were immobilized with 4% paraformaldehyde (Biosharp, Hefei, China) for 10 min, permeabilized with 0.05% Triton X-100 solution for 10 min at room temperature, and washed

with PBS. The sample was incubated for 30 min at room temperature in the dark with 200 µL of TRITC-labeled phalloidin working solution (Solarbio, Beijing, China). The sample was then counterstained with 200 µL DAPI (Solarbio, Beijing, China) for 30 s. Under the guidance of a laser scanning confocal microscope (Leica, Germany), cell samples were observed and photographed.

### Analysis of megakaryocyte differentiation and ploidy

Cells incubated with VLZ (2.5, 5, and 10 µM) for 5 days were collected and stained with FITC-CD41 and PE-CD42b (BioLegend, San Diego, CA, USA) antibodies for 15 min on ice in the dark. The percentages of $CD41^+/CD42b^+$ cells were evaluated by a flow cytometer (BD Biosciences, San Jose, CA, USA). Cells were fixed with 70% ice-cold ethanol for an overnight period. The samples were washed twice with PBS. The sample was then treated with 20 µg/mL propidium iodide (PI) and incubated on ice in the dark for 30 min. Finally, the cells were examined by flow cytometry. Cell apoptosis was measured by flow cytometry using the Annexin V-FITC/PI Apoptosis Detection Kit.

### Zebrafish model

The Chinese National Zebrafish Resource Center (Wuhan, China) provided transgenic zebrafish (itga2b: eGFP). All zebrafish procedures were conducted according to the guidelines of Southwest Medical University's laboratory animal ethics committee (License NO.20230716–002). The larvae of transgenic zebrafish expressing Tg (itga2b: eGFP) were collected at 3 days post fertilization (dpf) and distributed evenly in 12-well plates, with 20 larvae per plate. These larvae were subsequently exposed to varying concentrations (10–40 µM) of VLZ. At 5 dpf, the zebrafish were fixed in 4% paraformaldehyde overnight at 4 °C. Following fixation, the zebrafish larvae were embedded in agarose on a petri dish for imaging purposes. The total cells of 5 dpf zebrafish treated with VLZ were extracted from the lysate and detected by flow cytometry.

### Radiation-induced thrombocytopenia mouse model

We purchased specific pathogen-free (SPF) Kunming (KM) mice from Da-shuo Biotechnology Limited in Chengdu, Sichuan, China. The mice were approximately 9 weeks old, weighed approximately 20 g, and were kept under regular circumstances (22 ± 2 °C, 55 ± 5% humidity, and a 12 hr light and dark cycle). The experimental animal ethics committee of Southwest Medical University was followed during all procedures on mice (License NO.20230716–002). Except for the control group, the other mice were given a total body X-ray dose of 4 Gy after a week of acclimatization to induce a mouse model of thrombocytopenia. Each group of mice randomly consisted of six males and six females: normal group, model group, TPO group (3000 U/kg), and VLZ treated group (2.5, 5, and 10 mg/kg). TPO and VLZ were injected intraperitoneally daily post-irradiation for 12 days. Equal volumes of normal saline was intraperitoneally administered to the control and model groups.

### Hematologic parameters examination

On the indicated days after injection, blood (40 µL) was collected from the orbit and gently mixed with diluent (160 µL) to prevent clotting. The samples were measured by a Sysmex XT-2000iV automatic hematology analyzer (Kobe, Japan), including platelet counts (PLT), platelet volume (MPV), red blood cell counts (RBC), and white blood cell counts (WBC).

### Flow cytometry analysis of cells in vivo

Flow cytometry analysis of cells in vivo were performed as described previously (*Zhang et al., 2023*). Briefly, PB, BM, and spleen cells were examined by flow cytometry using the BD FACSCanto II flow cytometer (BD Biosciences, San Jose, CA, USA). The following antibodies were used: anti-CD41 (BioLegend, San Diego, CA, USA), anti-CD117 (c-Kit, BioLegend, San Diego, CA, USA), anti-61 (BD Biosciences, San Jose, CA, USA), PI/RNase Staining Buffer (BD Biosciences, San Jose, CA, USA), and anti-CD62P (BioLegend, San Diego, CA, USA). The antibodies were incubated in the dark for 30 min on ice.

### Carotid artery thrombosis test

The mice were administered normal saline, TPO, or VLZ daily post-irradiation for 12 days and then subjected to the carotid artery thrombosis test following a published protocol (*Xu et al., 2017*). In a

nutshell, filter paper (1 mm×3 mm) saturated with 10% $FeCl_3$ was placed on the left carotid artery for 3 min to induce thrombosis in the artery. The carotid artery was cleaned with PBS after the filter paper was removed. From the time of damage to stable occlusion (defined as no flow for 120 min), blood flow was constantly monitored with a vascular flow probe and Transonic TS420 flowmeter (Transonic Systems, Ithaca, NY, USA).

## Platelet aggregation

Blood was collected from the inferior vena cava of the mice after they had been given general anesthesia and placed in a syringe containing anticoagulant as previously described (*Lin et al., 2021*). After centrifugation (100×g) for 10 min to obtain platelet-rich plasma (PRP) and adjusted the concentration to $3×10^9$ platelets/mL by saline solution. ADP (Helena Laboratories, Beaumont, TX, USA) was used to stimulate the platelets, and platelet aggregation was measured using a turbidimetric aggregation-monitoring device (Helena Laboratories, Beaumont, TX, USA).

## Platelet adhesion

Each confocal dish was firstly filled with 1 mL of 5 g/mL collagen (Helena Laboratories, Beaumont, TX, USA) and left at 4 °C overnight. Each well was cleaned with PBS before being blocked for 1 hr at room temperature with 1% BSA. Then, PRP ($2×10^9$ cells/mL) was added, and the cells were incubated for 1 hr at 37 °C. After unbound platelets were removed with PBS, the rest of the steps were similar to the phalloidin staining, and TRITC-conjugated phalloidin was used to stain the adhering platelets. Representative images were acquired under a fluorescence microscope. Image J software was used to calculate the average cover of adhering platelets for quantification and statistical analysis (*Zhou et al., 2023*).

## Whole blood clot and washed platelet retraction

Whole blood was diluted with saline at a 1:3 ratio in a siliconized flat-bottomed glass test tube. Thrombin (Solarbio, Beijing, China) and $CaCl_2$ were used for the initial clotting and incubation at 37 °C for 2 hr was performed. After that, the supernatant was eliminated, and clots were weighed. Washed platelets were obtained as previously described. Washed platelets were diluted to $2×10^8$ platelets/mL with saline in glass test tubes. Then, the retraction was initiated with thrombin and $CaCl_2$. The test tubes were incubated at 37 °C for 2 hr. The percentage size of retracted platelets over time was determined from images using ImageJ (*Gauer et al., 2023*).

## Network pharmacology and molecular docking simulation

Network pharmacology and molecular docking simulation were performed according to a previously described protocol (*Ruan et al., 2023*). The overlapping targets of VLZ action and thrombocytopenia were obtained from the *SwissTargetPrediction, DisGeNET,* and *GeneCards* databases. For visibility and topology analysis, the eligible data were entered into *Cytoscape_v3.7.1* software using the STRING_v11.5 database. GO and KEGG analysis visualization using the *DAVID_v6.8* database, programs *Cytoscape/ClueGO_v3.9.0,* and *GraphPad Prism v9.1.1.233*. The crystal structures of the core target proteins, including MAPK1 (PDB:2Y9Q) and SRC (PDB:1O4A), were obtained from the *UniProt* website. These proteins were structurally altered using the *Sybyl-X 2.0* program. The total Surflex-Dock scores represent the binding affinities. Finally, the docking result was visualized utilizing *PyMOL_v2.5.2* and *Ligplus_v2.2 software* (*Rosignoli and Paiardini, 2022*; *Liu et al., 2022*).

## Western blot analysis

Total cellular proteins were harvested by 1x RIPA lysis buffer (Cell Signaling Technology, MA, USA) containing phosphatase inhibitor and protease inhibitor. The proteins were loaded and separated by SDS–PAGE, followed by transfer to a polyvinylidene difluoride (PVDF) membrane. The PVDF membrane was probed with the primary antibodies overnight at 4 °C. The blots were then incubated with secondary antibodies after rinsing three times with PBST. The blots were detected using an ECL detection reagent (4 A Biotech Co., Ltd., Beijing, China) and visualized with a scanner. Then quantified the bands were performed with ImageJ software. Positive signals were detected when the relative intensity of target proteins or phosphorylated proteins was compared with that of GAPDH or total proteins.

## Immunofluorescence staining

We collected cells treated with or without VLZ for 5 days and transferred them to microscope slides. These cells were fixed with 4% paraformaldehyde for 10 min, permeabilized with 0.1% Triton X-100, incubated with 1% BSA for 1 hr at room temperature, and incubated with primary antibodies for an additional 6 hr at 4 °C followed by the corresponding secondary antibody for 1 hr. DAPI was used to counterstain the nuclei for 10 min at room temperature in the dark. Images were taken under a fluorescence microscope.

## Drug affinity responsive target stability assay

Meg-01 cells were harvested and lysed with RIPA lysis buffer for 15 min on ice. The samples were centrifuged at 12,000 rpm for 15 min to obtain the supernatant. The protein concentration was measured with Bradford reagent and diluted to 5 mg/mL, followed by treatment with VLZ or DMSO for 1 hr at room temperature. Next, various ratios of pronase (1:500, 1:1000, and 1:1500) were added and incubated for 10 min at 40 °C. Then, 5x loading buffer was added to the sample and boiled at 95 °C for 10 min. The sample proteins were analyzed by Western blot.

## Statistical analysis

The mean and standard deviation (SD) of at least three independent experiments are used to express all data. One-way analysis of variance (ANOVA) and Tukey's post hoc test were used to evaluate the statistical comparisons among several groups. Two-tailed Student's $t$-tests were used to compare the differences between the two groups. Statistical significance was defined as a p-value <0.05.

## Acknowledgements

This work was supported by grants from the National Natural Science Foundation of China (82074129, 82374073 and 82004073), Sichuan Science and Technology Program (2022ZYD0087, 2022YFS0607, 22ZYZYTS0191, 2022YFS0614 and 2022JDJQ0061), Joint Project of Luzhou Municipal People's Government and Southwest Medical University (2020LZXNYDZ03 and 2020LZXNYDP01).

## Additional information

### Funding

| Funder | Grant reference number | Author |
| --- | --- | --- |
| National Natural Science Foundation of China | 82074129 | Jianming Wu |
| National Natural Science Foundation of China | 82374073 | Jianming Wu |
| National Natural Science Foundation of China | 82004073 | Feihong Huang |
| Natural Science Foundation of Sichuan Province | 2022ZYD0087 | Jianming Wu |
| Natural Science Foundation of Sichuan Province | 2022YFS0607 | Jianming Wu |
| Natural Science Foundation of Sichuan Province | 22ZYZYTS0191 | Jianming Wu |
| Natural Science Foundation of Sichuan Province | 2022YFS0614 | Yiwei Wang |
| Natural Science Foundation of Sichuan Province | 2022JDJQ0061 | Jianming Wu |

| Funder | Grant reference number | Author |
|---|---|---|
| Science and Technology Planning Project of Luzhou City | 2020LZXNYDZ03 | Jianming Wu |
| Science and Technology Planning Project of Luzhou City | 2020LZXNYDP01 | Jianming Wu |

The funders had no role in study design, data collection and interpretation, or the decision to submit the work for publication.

## Author contributions

Ling Zhou, Conceptualization, Investigation, Writing - original draft; Chengyang Ni, Investigation, Writing - original draft; Ruixue Liao, Taian Yi, Investigation, Methodology; Xiaoqin Tang, Validation, Investigation; Mei Ran, Data curation, Investigation; Miao Huang, Rui Liao, Software, Investigation; Xiaogang Zhou, Supervision, Validation, Investigation; Dalian Qin, Supervision; Long Wang, Supervision, Validation; Feihong Huang, Resources, Supervision; Xiang Xie, Resources, Software; Ying Wan, Resources; Jiesi Luo, Conceptualization, Software, Supervision, Writing – review and editing; Yiwei Wang, Conceptualization, Software, Supervision, Funding acquisition, Writing – review and editing; Jianming Wu, Conceptualization, Supervision, Funding acquisition, Writing – review and editing

## Author ORCIDs

Ling Zhou ⬤ http://orcid.org/0000-0001-8620-5557
Jianming Wu ⬤ http://orcid.org/0000-0002-6136-7469

## Ethics

This study was performed in strict accordance with the recommendations in the Guide for the Care and Use of Laboratory Animals of Southwest Medical University. All zebrafish and mouse procedures were conducted according to the guidelines of Southwest Medical University's laboratory animal ethics committee (License NO.20230716-002).

Reviewer #1 (Public review): https://doi.org/10.7554/eLife.94765.3.sa1
Reviewer #2 (Public review): https://doi.org/10.7554/eLife.94765.3.sa2
Author response https://doi.org/10.7554/eLife.94765.3.sa3

# Additional files

## Supplementary files

• MDAR checklist

## Data availability

The dataset is available in Dryad: https://doi.org/10.5061/dryad.7m0cfxq34.

The following dataset was generated:

| Author(s) | Year | Dataset title | Dataset URL | Database and Identifier |
|---|---|---|---|---|
| Zhou L, Ni C, Liao R, Tang Y, Ran M, Huang M, Liao R, Zhou X, Qin D, Wang L, Huang F, Xie X, Wan Y, Luo J, Wang Y, Wu J | 2024 | Activating SRC/MAPK signaling via 5-HT1A receptor contributes to the effect of vilazodone on improving thrombocytopenia | https://doi.org/10.5061/dryad.7m0cfxq34 | Dryad Digital Repository, 10.5061/dryad.7m0cfxq34 |

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
