## [Editor Report · eLife assessment]

This study presents a rather **valuable** finding that vilazodone can restore the normal platelet level through regulating 5-HT1A receptor. The evidence supporting the claims of the authors is **solid**, although inclusion of more cell lines and more detailed analysis of the results would have strengthened the study. The work will be of interest to scientists working in the field of thrombocytopenia.

---

## [Referee Report · Reviewer #1 (Public review)]

Summary:

This is well-performed research with solid results and thorough control. The authors did a good job of finding the relationship between the 5-HT1A receptor and megakaryocytopoiesis, which demonstrated the potential of vilazodone in the management of thrombocytopenia. It emphasizes the regulatory mechanism of 5-HT1A receptor signaling on hematopoietic lineages, which could further advance the field of thrombocytopenia for therapeutic purposes.

Strengths:

This is a comprehensive and detailed research using multiple methods and model systems to determine the pharmacological effects and molecular mechanisms of vilazodone. The authors conducted in vitro experiments using HEL and Meg-01 cells and in vivo experiments using Zebrafish and Kunming-irradiated mice. The experiments and bioinformatics analysis have been performed with a high degree of technical proficiency. The authors demonstrated how vilazodone binds to 5-HTR1A and regulates the SRC/MAPK pathway, which is inhibited by particular 5-HTR1A inhibitors. The authors determined this to be the mechanistic underpinning for the effects of vilazodone in promoting megakaryocyte differentiation and thrombopoiesis.

Weaknesses:

(1) Which database are the drug test sets and training sets for the creation of drug screening models obtained from? What criteria are used to grade the results?

(2) What is the base of each group in Figure 3b for the survival screening of zebrafish? The positivity rate of GFP-labeled platelets is too low, as indicated by the quantity of eGFP+ cells. What gating technique was used in Figure 3e?

(3) In Figure 4C, the MPV values of each group of mice did not show significant downregulation or upregulation. Please explain the possible reasons.

(4) The PPI diagram and the KEGG diagram in Figure 6 both provide a possible mechanism pathway for the anti-thrombocytopenia effect of vilazodone. How can the author analyze the differences in their results?

(5) 5-HTR1A protein expression is measured only in the Meg-01 cells assay. Similar quantitation through western blot is not shown in other cell models.

---

## [Referee Report · Reviewer #2 (Public review)]

Summary:

The authors tried to understand the mechanism on how a drug candidate, VLZ, works on a receptor, 5-HTR1A, by activating the SRC/MAPK pathway to promote the formation of platelets.

Strengths:

The authors used both computational and experimental methods. This definitely saves time and funds to find a useful drug candidate and its therapeutic marker in the subfield of platelets reduction in cancer patients. The authors achieved the aim to explain the mechanism of VLZ on improving thrombocytopenia by using two cell lines and two animal models.

Weaknesses:

Only two cell lines, HEL and Meg-01 cells, were evaluated in this study. However, using more cell lines is really depending on the work flow and the grant situations of the current research team.

---

## [Author Response]

The following is the authors’ response to the original reviews.

**eLife assessment:**
This study presents a valuable finding on the possible use of vilazodone in the management of thrombocytopenia through regulating 5-HT1A receptor signaling. The evidence supporting the claims of the authors is solid, with the combined use of computational methods and biochemical assays. The work will be of broad interest to scientists working in the field of thrombocytopenia.
**Public Review:**

**Reviewer #1 (Public Review):**
Summary:This is well-performed research with solid results and thorough controls. The authors did a good job of finding the relationship between the 5-HT1A receptor and megakaryocytopoiesis, which demonstrated the potential of vilazodone in the management of thrombocytopenia. The paper emphasizes the regulatory mechanism of 5-HT1A receptor signaling on hematopoietic lineages, which could further advance the field of thrombocytopenia for therapeutic purposes.Strengths:This is comprehensive and detailed research using multiple methods and model systems to determine the pharmacological effects and molecular mechanisms of vilazodone. The authors conducted in vitro experiments using HEL and Meg-01 cells and in vivo experiments using Zebrafish and Kunming-irradiated mice. The experiments and bioinformatics analysis have been performed with a high degree of technical proficiency. The authors demonstrated how vilazodone binds to 5-HTR1A and regulates the SRC/MAPK pathway, which is inhibited by particular 5-HTR1A inhibitors. The authors determined this to be the mechanistic underpinning for the effects of vilazodone in promoting megakaryocyte differentiation and thrombopoiesis.Weaknesses:(1) Which database are the drug test sets and training sets for the creation of drug screening models obtained from? What criteria are used to grade the results?

Response: Thank you for your thoughtful comment. The database is built by our laboratory. Firstly, we collected 39 small molecule compounds that can promote MK differentiation or platelet formation and 691 small molecule compounds that have no obvious effect on MK differentiation or platelet formation to buiid the datbase. Then, the data of the remaining 713 types of small molecule compounds were utilized as the Training set, and the Molecular Descriptors of 2 types of active and 15 types of inactive small molecule compounds were randomly picked as the Validation set.With regard to the activity evaluation criteria, the prediction score for each molecule was between 0 and 1, and the model decision was made with a threshold of 0.5. The molecule with a score above the 0.5 threshold was identified as a megakaryopoiesis inducer (1).

Reference：

(1) Mo Q, Zhang T, Wu J, et al. Identification of thrombopoiesis inducer based on a hybrid deep neural network model. Thromb Res. 2023;226:36-50. doi:10.1016/j.thromres.2023.04.011

(2) What is the base of each group in Figure 3b for the survival screening of zebrafish? The positivity rate of GFP-labeled platelets is too low, as indicated by the quantity of eGFP+ cells. What gating technique was used in Figure 3e?

Response: We are deeply grateful for the insightful feedback you have provided regarding Figure 3 and the assessment of zebrafish model. We used 50 zebrafish embryos per group to evaluate VLZ toxicity, and we think this is a suitable and fair baseline. Our gating procedure is clearly depicted in the resulting diagram. Since our goal was to evaluate the fluorescence intensity quantitatively, we isolated the entire zebrafish cell. Since the amount of eGFP+ in various zebrafish tissues found in other literature is likewise quite low and we are unsure of the typical eGFP+ threshold for zebrafish (1, 2), we think this finding should be fair given that each group's activities in the experiment were conducted in parallel.

Reference：

(1) Yang L, Wu L, Meng P, et al. Generation of a thrombopoietin-deficient thrombocytopenia model in zebrafish. J Thromb Haemost. 2022; 20(8): 1900-1909. doi:10.1111/jth.15772

(2) Fallatah W, De Silva IW, Verbeck GF, Jagadeeswaran P. Generation of transgenic zebrafish with 2 populations of RFP- and GFP-labeled thrombocytes: analysis of their lipids. Blood Adv. 2019;3(9):1406-1415. doi:10.1182/bloodadvances.2018023960

(3) In Figure 4C, the MPV values of each group of mice did not show significant downregulation or upregulation. The possible reasons for this should be explained.

Response: Thank you for your thoughtful comment. Megakaryocytes build pseudopodia, which form extensions that release proplatelets into the bone marrow sinusoids. Proplatelets convert into barbell-shaped proplatelets to form platelets in an integrin αIIbβIII mediated process (1-2). Platelet size is established by microtubule and actin-myosin-sceptrin cortical forces which determine platelet size during the vascular formation of barbell proplatelets (3). Conversion is regulated by the diameter and thickness of the peripheral microtubule coil. Proplatelets can also be formed from proplatelets in the circulation (4). Megakaryocyte ploidy correlates with platelet volume following a direct nonlinear relationship to mean platelet volumes (5). Usually there is an equilibrium between platelet generation and clearance from the circulation (normal turnover) controlled by thrombopoietin. When healthy humans receive thrombopoietin, their platelet size decreases (6). Proplatelet formation is dynamic and influenced by platelet turnover (7) which increases upon increased platelet consumption and/or sequestration. In our study, the MPV values of each group of mice did not show significant downregulation or upregulation, from our point of view, there are several possible reasons for these results.

(1) Mice in a radiation-damaged state may result in a decrease in platelet count, but at the same time stimulate the bone marrow to release young and larger platelets, thus keeping the MPV relatively stable.

(2) After radiation injury, bone marrow cells were suppressed, resulting in a decrease in the number of platelets produced, but MPV remained unchanged, possibly because the direct effects of radiation on the bone marrow caused thrombocytopenia, but not necessarily the average platelet size.

Reference：

(1) Thon JN, Italiano JE. Platelet formation. Semin Hematol. 2010(3):220-226. doi: 10.1053/j.seminhematol.2010.03.005.

(2) Larson MK, Watson SP. Regulation of proplatelet formation and platelet release by integrin alpha IIb beta3. Blood. 2006(5):1509-1514. doi: 10.1182/blood-2005-11-011957.

(3) Thon JN, Macleod H, Begonja AJ, et al., Microtubule and cortical forces determine platelet size during vascular platelet production. Nat. Commun. 2012(3):852. doi: 10.1038/ncomms1838.

(4) Machlus KR, Thon JN, Italiano JE Jr. Interpreting the developmental dance of the megakaryocyte: a review of the cellular and molecular processes mediating platelet formation. Br. J. Haematol. 2014(2):227-36. doi: 10.1111/bjh.12758.

(5) Bessman JD. The relation of megakaryocyte ploidy to platelet volume. Am. J. Hematol. 1984(2):161-170. doi: 10.1002/ajh.2830160208.

(6) Harker LA, Roskos LK, Marzec UM, et al., Effects of megakaryocyte growth and development factor on platelet production, platelet life span, and platelet function in healthy human volunteers. Blood. 2000(8):2514-2522. doi: 10.1182/blood.V95.8.2514.

(7) Kowata S, Isogai S, Murai K, et al., Platelet demand modulates the type of intravascular protrusion of megakaryocytes in bone marrow. Thromb. Haemost. 2014(4):743-756. doi: 10.1160/TH14-02-0123.

(4) The PPI diagram and the KEGG diagram in Figure 6 both provide a possible mechanism pathway for the anti-thrombocytopenia effect of vilazodone. How can the authors analyze the differences in their results?

Response: We are appreciated your valuable comments. PPI (Protein-Protein Interaction) refers to the interaction between proteins. Inside cells, proteins interact with each other to perform various biological functions, influencing cell signaling, metabolic pathways, cell cycle, and more. KEGG (Kyoto Encyclopedia of Genes and Genomes) is a database that integrates information on genomes, chemicals, and biological systems. In pharmacoinformatic, KEGG pathways are often used to understand the molecular mechanisms of specific diseases or biological processes. KEGG contains the interrelationships between genes, proteins, and metabolites, helping to reveal key nodes in biological processes. PPI information can be integrated with data from KEGG pathways, such as metabolic and signaling pathways, to gain a more comprehensive understanding of the role of protein-protein interactions in cellular processes and biological functions. For example, by analyzing nodes in the PPI network, proteins associated with a specific disease can be identified, and further examination of these proteins' locations in KEGG pathways can reveal molecular mechanisms underlying the onset and development of the disease. However, this method also has some limitations：

Uncertainty (1): The construction of protein-protein interaction networks and drug interaction networks involves many assumptions and speculations. The edges of these networks may be based on experimental data but can also rely on bioinformatics predictions. Therefore, the accuracy of predictions is limited by the quality and reliability of the data used during network construction.

Insufficient data (2): Despite the availability of a large amount of bioinformatics data for network construction, interactions between some proteins and drugs may still lack sufficient experimental data. This data insufficiency can result in inaccuracies in network predictions.

Dynamics and temporal-spatial changes (3): The dynamics and temporal-spatial changes in biological systems are crucial for drug effects. Pharmacoinformatic may struggle to capture these changes as it often relies on static network representations, overlooking the temporal and dynamic nature of biological systems.

Reference：

(1) Fernando PC, Mabee PM, Zeng E. Integration of anatomy ontology data with protein-protein interaction networks improves the candidate gene prediction accuracy for anatomical entities. BMC Bioinformatics. 2020(1):442. doi: 10.1186/s12859-020-03773-2.

(2) Zhang S, Zhao H, Ng MK. Functional module analysis for gene coexpression networks with network integration. IEEE/ACM Trans. Comput. Biol. Bioinform. 2015(5):1146-1160. doi: 10.1109/TCBB.2015.2396073.

(3) Cinaglia P, Cannataro M. A method based on temporal embedding for the pairwise alignment of dynamic networks. Entropy (Basel). 2023(4):665. doi: 10.3390/e25040665.

(5)-HTR1A protein expression is measured only in the Meg-01 cells assay. Similar quantitation through western blot is not shown in other cell models.

Response: Your insightful criticism and recommendation to use different cell models in order to obtain a more accurate depiction of 5-HTR1A protein expression are greatly appreciated. We completely concur that using this strategy would greatly increase the validity of our research. However, establishing a primary megakaryocyte model requires specialized expertise and technical resources, which unfortunately are not readily available to us within the given timeframe. Nevertheless, we acknowledge the limitations of Meg-01 cells, which may exhibit distinct properties compared to true megakaryocytes. To mitigate this concern, we have ensured robust experimental design and rigorous data analysis to interpret our findings within the context of these model cell lines. We believe our results still provide valuable insights into megakaryocyte differentiation and address an important biological question.

**Reviewer #2 (Public Review):**
Summary:The authors tried to understand the mechanism of how a drug candidate, VLZ, works on a receptor, 5-HTR1A, by activating the SRC/MAPK pathway to promote the formation of platelets.Strengths:The authors used both computational and experimental methods. This definitely saves time and funds to find a useful drug candidate and its therapeutic marker in the subfield of platelets reduction in cancer patients. The authors achieved the aim of explaining the mechanism of VLZ in improving thrombocytopenia by using two cell lines and two animal models.Weaknesses:Only two cell lines, HEL and Meg-01 cells, were evaluated in this study. However, using more cell lines is really depending on the workflow and the grant situations of the current research team.

Response: We deeply appreciate your insightful feedback and valuable suggestions regarding the use of more suitable models for studying the role of VLZ in megakaryocyte differentiation and platelet production. We fully agree that CD34+ hematopoietic stem/progenitor cells or primary megakaryocytes would provide a more accurate representation of in vitro megakaryopoiesis compared to HEL and Meg-01 cells, which possess limited potential for this process. We acknowledge that our current study did not include experiments with these preferred cell models. This is because our laboratory is still actively developing the technical expertise and resources required for establishing and maintaining primary megakaryocyte and CD34+ cell cultures. Despite the limitations of the current study, we believe the results using HEL and Meg-01 cells provide valuable preliminary insights into the potential effects of VLZ on megakaryocyte differentiation. We are actively working to overcome these limitations and plan to incorporate these more advanced models in our future investigations.

**Reviewer #1 (Recommendations For The Authors):**
I think the authors can enhance the mechanism study by developing more reliable models and methodologies. The connection to clinical research should be strengthened at the same time.

Response: We deeply appreciate your insightful feedback and valuable suggestions regarding the use of more suitable models for studying the role of VLZ in megakaryocyte differentiation and platelet production. Despite the limitations, we are committed to expanding our research in the future by incorporating your suggestion and establishing a primary megakaryocyte model to further validate our findings and strengthen our conclusions. At the same time, we wholeheartedly concur with your suggestion to combine clinical research. Unfortunately, VLZ is not a first-line treatment for depression in China, and getting blood samples from the matching number of patients for analysis is a challenge. To give additional experimental support for the medication, we have attempted to improve the data in vivo as much as feasible, including by implementing the intervention in normal mice. Our findings should also contribute to the theoretical underpinnings of this medication and aid in its practical application.

**Reviewer #2 (Recommendations For The Authors):**
Issues the authors need to address:Figure 7: Why the band intensity of GAPDH in b or e is much greater than that in f, g, or h?

Response: Thank you for your careful observation and insightful comment regarding Figure 7. Because the concentration of each batch of protein samples is different, sometimes the GAPDH band strength is increased by the large loading volume. Other factors that may influence the GAPDH band strength include the instrument's contrast adjustment during exposure and the use of different numbers of holes for electrophoresis. Meanwhile, the original three replicate results of all WB results will be provided in the supplementary materials.

Finally, we sincerely thank you for providing us with this opportunity to make a further revision and modification of our manuscript, and your valuable and scientific comments are useful for the great improvement of our manuscript!